# 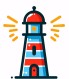 BEACON: Benchmark for Comprehensive RNA Tasks and Language Models

**Yuchen Ren**[1,2] *    **Zhiyuan Chen**[1,3] *    **Lifeng Qiao**[1,4] *
**Hongtai Jing**[5]    **Yuchen Cai**[1,5]    **Sheng Xu**[1,5]    **Peng Ye**[1,5 †, #]    **Xinzhu Ma**[1,6 †]
**Siqi Sun**[1,5]    **Hongliang Yan**[1]    **Dong Yuan**[2]    **Wanli Ouyang**[1]    **Xihui Liu**[1,3]
*equal contribution    †corresponding author    #project leader
[1]Shanghai AI Lab    [2]USYD    [3]HKU    [4]SJTU    [5]FDU    [6]CUHK
{renyuchen, chenzhiyuan, qiaolifeng, caiyuchen, xusheng, yepeng, maxinzhu, sunsiqi, yanhongliang,
ouyangwanli, liuxihui}@pjlab.org.cn, htjing21@m.fudan.edu.cn, dong.yuan@sydney.edu.au

## Abstract

RNA plays a pivotal role in translating genetic instructions into functional outcomes, underscoring its importance in biological processes and disease mechanisms. Despite the emergence of numerous deep learning approaches for RNA, particularly universal RNA language models, there remains a significant lack of standardized benchmarks to assess the effectiveness of these methods. In this study, we introduce the first comprehensive RNA benchmark BEACON (**BE**nchm**A**rk for **CO**mprehensive R**N**A Task and Language Models). First, BEACON comprises 13 distinct tasks derived from extensive previous work covering structural analysis, functional studies, and engineering applications, enabling a comprehensive assessment of the performance of methods on various RNA understanding tasks. Second, we examine a range of models, including traditional approaches like CNNs, as well as advanced RNA foundation models based on language models, offering valuable insights into the task-specific performances of these models. Third, we investigate the vital RNA language model components from the tokenizer and positional encoding aspects. Notably, our findings emphasize the superiority of single nucleotide tokenization and the effectiveness of Attention with Linear Biases (ALiBi) over traditional positional encoding methods. Based on these insights, a simple yet strong baseline called BEACON-B is proposed, which can achieve outstanding performance with limited data and computational resources. The datasets and source code of our benchmark are available at `https://github.com/terry-r123/RNABenchmark`.

## 1 Introduction

RNA plays a vital role in numerous biological processes, including protein synthesis, enzymatic activities, and gene regulations [22, 73, 115, 85]. Unlike its more famous counterpart DNA, RNA is not restricted to information storage but actively participates in translating genetic instructions into functional proteins, modulating gene expression through various mechanisms, and regulating cellular responses to internal and external stimuli [39]. As a dynamic intermediary between DNA and protein, RNA governs crucial biological processes, making it a focal point of research in molecular biology and biomedicine. Consequently, understanding the diverse functions of RNA is crucial to unraveling the complexities of cellular processes and deciphering the underlying mechanisms of diseases.

Despite its critical importance, understanding the functional roles of RNA poses significant challenges. Inspired by the success of machine learning in various fields, there have been extensive research efforts in recent years to apply machine learning approaches to RNA tasks. Initially, traditional

machine learning algorithms such as support vector machine and random forest paved the way for predictive modeling in RNA studies [58, 101, 63]. The evolution of deep learning, especially through Convolutional Neural Networks (CNNs), has enabled more nuanced analyses of RNA sequences and structures [89, 9, 53]. More recently, pre-trained language models (LM) have revolutionized RNA research, facilitating more accurate predictions of RNA function and interactions [13, 18, 119]. These advancements significantly deepen our understanding of RNA's regulatory roles in cellular processes.

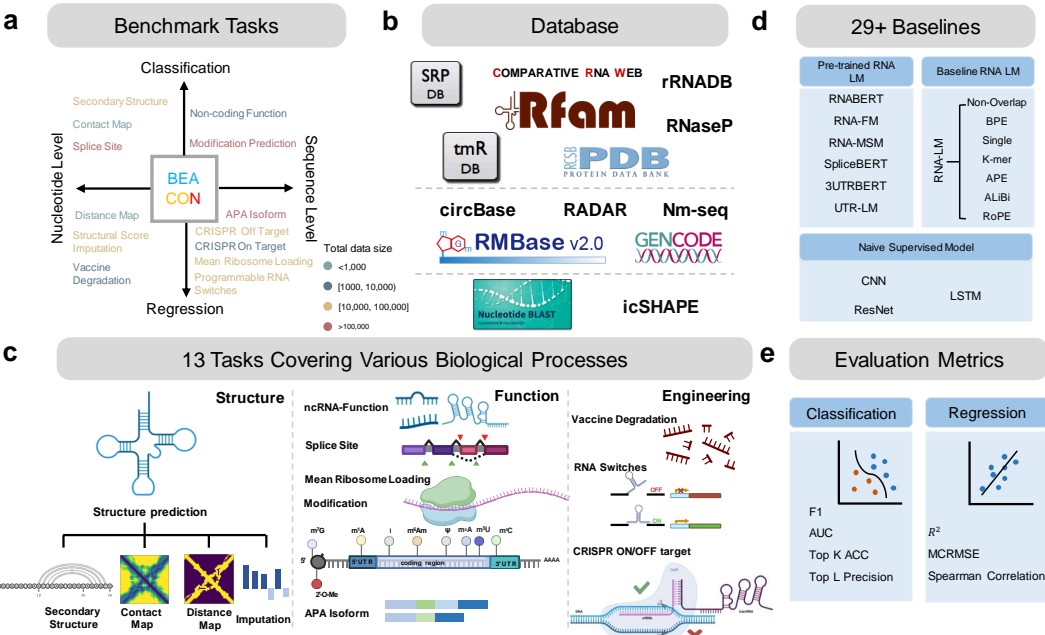

Figure 1: Overview of BEACON. **a:** Categorization of the 13 benchmark tasks into classification and regression at both nucleotide and sequence levels. **b:** Diverse database distinguished by data size and source type. **c:** Visual representations of tasks across Structure, Function, and Engineering. **d:** List of baseline models, including naive supervised deep models and advanced RNA language models. **e:** Metrics for evaluating model performance in classification and regression tasks, tailored to RNA analysis specifics.

According to the central dogma of molecular biology [20], genetic information flows unidirectionally from DNA to RNA and then to protein, or directly from RNA to protein. While established benchmarks for DNA [41, 68] and protein [78, 117] have significantly aided research in these areas, RNA, a crucial component of the central dogma, lacks such standardized benchmarks. Therefore existing RNA models are often evaluated using disparate individual datasets, making it difficult to conduct fair comparisons between different methods and hindering the development of the field.

To address this gap, we present the first comprehensive RNA benchmark called BEA-CON (**BE**nchm**A**rk for **CO**mprehensive R**N**A Task and Language Models). As shown in Fig. 1, BEACON contains a curated collection of 13 important RNA-related tasks derived from a comprehensive review of RNA-related research papers [24, 53, 19], containing 967k sequences with lengths ranging from 23 to 1182. These tasks cover sequence-level and nucleotide-level analyses across three main fields: **Structural Analysis** focuses on deciphering RNA's secondary structures and three-dimensional configurations, essential for its interactions with other molecules and for therapeutic design. **Functional Studies** investigate RNA's roles in gene regulation and its implications for disease, which are vital for protein translation and treatment of disease. **Engineering Applications** explore RNA's potential in synthetic biology to enhance its utility in biotechnology and medicine, exploring how RNA can be utilized to solve complex biological challenges.

In addition, we evaluate a diverse range of models using our benchmark, including traditional models like CNNs, ResNets, and LSTMs, as well as advanced RNA foundation models like RNA-FM [13] and UTR-LM [18]. Surprisingly, ResNet and LSTM are proved to be strong baselines, managing to outperform language models on several tasks. Additionally, pretrained RNA language models surpassed previous task-specific state-of-the-art (SOTA) performances on 8 out of the 13 tasks,

demonstrating significant potential. Next, we explore the impact of various components in RNA language models for the community, with a particular focus on tokenization methods and positional encodings. We conclude some experimental findings, based on which we further propose a robust yet efficient baseline, BEACON-B, that incorporates Attention with Linear Biases (ALiBi) and single nucleotide tokenization, providing an extremely fast and easy-to-use open-source pre-training model for the community.

Overall, our contributions can be summarized as follows:

- We establish the first comprehensive benchmark for RNA research with 13 diverse tasks, covering structure, function, and engineering aspects.
- We conduct a thorough evaluation of pre-trained RNA language models, providing insights into their strengths and limitations across different tasks.
- We investigate the component impacts of RNA language models in depth, and propose BEACON-B, a simple yet strong baseline, that benefits subsequent research in the field.

## 2 Related Works

**RNA tasks.** RNA research is categorized into three primary areas: structure, function, and engineering. Structural tasks, such as predicting secondary structures [24] and contact maps [102], aim to understand RNA configurations. Functional tasks focus on the biological roles of RNA, including splice site prediction [53] and non-coding RNA function classification [3]. Engineering tasks involve designing RNA molecules with specific properties for applications in synthetic biology, such as discriminating programmable RNA switches [7].

**Deep learning methods in RNA tasks.** Deep learning has been pivotal in addressing these tasks. For structural predictions, U-Net [83] has been employed to model secondary structures [36]. In functional studies, methods like SpliceAI utilize dilated convolutions for effective splice detection [53]. For engineering challenges, LSTMs have been used to design programmable RNA switches, demonstrating their versatility in handling complex sequence data [7]. The development of foundational RNA models like RNA-FM [13], RNA-BERT [1], RNA-MSM [124], SpliceBERT [15], UTR-LM [18] and 3UTRBERT [119] represents a significant advancement, capable of tackling multiple RNA-related tasks by leveraging advanced language modeling techniques. These models promise a broader understanding of RNA biology. However, they often lack thorough evaluations across different tasks, highlighting a gap in the systematic assessment of their capabilities. For instance, UTR-LM [18] only focuses on 5' UTR function-related tasks. This limitation underscores the need for robust, cross-disciplinary evaluation approaches to fully explore and utilize the potential of these models in RNA research.

**Benchmarks in molecular biology.** While AI for RNA research is a relatively new field and lacks comprehensive benchmarks, numerous benchmarks have been established for DNA [41, 23, 69] and protein [78, 117, 74, 38] studies. Grešováet al. proposed Genomic Benchmarks [41], which includes a collection of genomic sequence classification tasks. Marin et al. constructed BEND [69], a comprehensive benchmark for DNA, encompassing tasks such as gene finding, enhancer annotation, histone modification, CpG methylation, etc. Notin et al. introduced ProteinGym [74], a benchmark specifically designed for protein fitness prediction and design, while Gao et al. proposed ProteinBench [38], focusing on protein design. Additionally, Xu et al. developed PEER [117], a comprehensive protein benchmark involving function, localization, structure, etc. As AI for RNA research develops, some preliminary benchmarks in RNA have emerged. RnaBench [87] is the benchmark for computational RNA modeling, though it only involves tasks related to RNA secondary structure and RNA design. Many real-life applications, such as RNA-based therapeutics, require a comprehensive understanding of the functions of ncRNA and mRNA [126]. To address this gap, we developed BEACON, a comprehensive benchmark for RNA that covers a wide range of tasks covering structural analysis, functional studies, and engineering applications.

## 3 Benchmark Tasks

BEACON comprises 13 tasks designed to evaluate RNA models comprehensively, covering structural analysis, functional studies, and engineering applications . The following sections provide detailed information, including data statistics, evaluation metrics, and data sources as shown in Table 1.

## 3.1 Structure Prediction

**Secondary Structure Prediction (SSP)** identifies paired regions (stems) and unpaired regions (loops, bulges, junctions) within RNA molecules. The target matrix is $y \in \mathbb{R}^{l \times l}$ indicating whether each nucleotide pair forms a base pair as part of the RNA's secondary structure. We adopt the bpRNA-1m database [24], which contains detailed annotations of over 100,000 single-molecule RNA structures. The evaluation metric is the F1 score.

*Impact:* Accurate secondary structure prediction is pivotal for elucidating the structural and functional dynamics of RNA. By precisely mapping these structures, researchers gain insights into functional regions and interaction sites, contributing significantly to areas such as drug discovery and genetic research.

**Contact Map Prediction (CMP)** identifies pairs of nucleotides in RNA that are in close proximity in their three-dimensional structures. Each nucleotide pair is associated with a binary label $y \in \{0, 1\}$ indicating whether they contact (within a distance threshold of 8 Å). Following [102], we utilize a dataset derived from non-redundant RNA 3D structures documented by [61], and evaluate predictions using the Top-L precision metric.

*Impact:* Accurately identifying nucleotide interactions is pivotal for inferring the tertiary structure of RNA molecules. These predictions enhance our understanding of RNA folding and function, contributing to advancements in RNA-based therapeutics and biotechnology.

**Distance Map Prediction (DMP)** estimates the physical distances between pairs of nucleotides within an RNA molecule. The target distance matrix $y \in \mathbb{R}^{l \times l}$ records the distance between every pair of nucleotides within the sequence. The same dataset as the contact map prediction task is used, with $R^2$ serving as the evaluation metric.

*Impact:* Inter-nucleotide distance prediction offers detailed spatial information, facilitating the construction of accurate three-dimensional models by providing distance restraints.

**Structural Score Imputation (SSI)** predicts missing structural information within RNA molecules, with each nucleotide assigned with an experimentally derived structural score $y \in R$. The dataset [40] is derived from icSHAPE sequencing data of the HEK293 cell line, with 30% of nucleotides randomly masked as null in the training set, and downsampling leading to missing values in 3,095 fragments in the testing set. The evaluation metric is $R^2$ .

*Impact:* Accurate structural score imputation provides enhanced and comprehensive structural information, crucial for the development of RNA-based therapeutics and diagnostics. Improved structural data enable precise targeting of RNA molecules in disease treatment, potentially leading to more effective interventions.

## 3.2 Function Prediction

**Splice Site Prediction (SPL)** classifies each base within a sequence into one of three categories: acceptor (a), donor (d), or neither (n), with the categorical label $y \in \{0, 1, 2\}$. The task uses Jaganathan's dataset [53] with Top-k accuracy as the evaluation metric.

*Impact:* Splice site prediction is crucial for studying gene expression and regulation within biological systems. Accurate prediction of splice sites helps determine the precise locations within the genome where splicing occurs, enabling the detection of non-coding genomic variations that could impact protein synthesis, particularly those resulting in cryptic splicing.

**APA Isoform Prediction (APA)** predicts the usage ratio of the proximal polyadenylation site (PAS) in the 3' untranslated region (3' UTR) for each variant, recorded in target $y \in \mathbb{R}$. We filter 228k sequences from over 3 million APA reporter gene data from Bogard's dataset [9], this regression task assesses the proportion of proximal APA isoforms. The evaluation metric is the $R^2$ value.

*Impact:* APA is a common gene expression regulation mechanism that generates different RNA transcripts and protein isoforms by modulating RNA 3' UTR processing [121]. This regulatory method can affect gene expression levels and functions, playing a crucial role in cellular biological processes and development.

**Non-coding RNA Function Classification (ncRNA)** classifies ncRNA molecules into categories like microRNAs (miRNAs), long non-coding RNAs (lncRNAs), and small interfering RNAs (siRNAs).

Table 1: Overview of the 13 benchmark tasks across three major RNA task groups. Nucleotide-level tasks require labels to have the same length as input sequences. Sequence-level tasks require nucleotides from one input sequence to share one label. Cls and Reg denote classification and regression, respectively. MCRMSE means Mean Columnwise Root Mean Squared Error.

| RNA Task | #Train/Validation/Test | Metric | Task Type | RNA Level | Max/Mean Length | Source/Venue |
|---|---|---|---|---|---|---|
| | | | Structure | | | |
| SSP | 10,814/1,300/1,305 | F1 | Multi-label Cls | Nucleotide | 499/133.8 | bpRNA-1m/NC [95] |
| CMP | 188/23/80 | Top L Precision | Multi-label Cls | Nucleotide | 960/110.3 | RNAcontact/BIOINF [61] |
| DMP | 188/23/80 | $R^2$ | Reg | Nucleotide | 960/110.3 | RNAcontact/BIOINF [61] |
| SSI | 14,049/1,756/3,095 | $R^2$ | Reg | Nucleotide | 100/100 | StructureImpute/NMI [40] |
| | | | Function | | | |
| SPL | 144,628/18,078/16,505 | Top-k ACC | Multi-class Cls | Nucleotide | 100/100 | SpliceAI/Cell [53] |
| APA | 145,463/33,170/49,755 | $R^2$ | Reg | Sequence | 186/186 | APARENT/Cell [9] |
| ncRNA | 5,679/650/2,400 | ACC | Multi-class Cls | Sequence | 1182/158.4 | Noorul's/NMI [3] |
| Modif | 304,661/3,599/1,200 | AUC | Multi-label Cls | Sequence | 101/101 | MultiRM/NC [97] |
| MRL | 76,319/7,600/7,600 | $R^2$ | Reg | Sequence | 100/61.5 | Optimus/NBT [89] |
| | | | Engineering | | | |
| VDP | 2,155/245/629 | MCRMSE | Multi-label Reg | Nucleotide | 107/118.5 | OpenVaccine/NMI [111] |
| PRS | 73,227/9,153/9,154 | $R^2$ | Multi-label Reg | Sequence | 148/148 | Angenent-Mari's/NC [6] |
| CRI-On | 1,453/207/416 | Spearman Corr | Reg | Sequence | 23/23 | DeepCRISPR/GB [19] |
| CRI-Off | 14,223/2,032/4,064 | Spearman Corr | Reg | Sequence | 23/23 | DeepCRISPR/GB [19] |

Each molecule is assigned a categorical label $y \in \{0, 1, ..., 12\}$ to denote its function. The dataset [3, 34] comprises contributions from GENCODE, circBase, and Rfam, encompassing various ncRNAs. Accuracy (ACC) at the sequence level is the evaluation metric.

*Impact:* Classifying ncRNA functions is crucial for understanding their diverse roles in gene regulation and cellular processes. Accurate classification enhances our knowledge of regulatory networks and aids in elucidating disease mechanisms. This contributes to identifying new biomarkers and therapeutic targets, advancing molecular biology research, and improving disease diagnosis and treatment.

**Modification Prediction (Modif)** predicts twelve widely occurring types of RNA modifications from a given RNA sequence, indicated by a categorical label $y \in \{0, 1, ..., 11\}$. We adopt Song's dataset that contains 20 epi-transcriptome profiles for 12 different types of RNA modifications obtained from 15 base-resolution technologies, where over 300,000 sites were collected and divided into training, validation, and test sets. We use AUC as the metric.

*Impact:* Post-transcriptional RNA modifications enhance the structural and functional diversity of RNA molecules, impacting all stages of RNA life [32]. Due to the complex and diverse characteristics of RNA sequences, different modifications may correspond to distinct sequence features. Accurately identifying RNA modification sites is crucial for understanding the functions and regulatory mechanisms of various RNAs.

**Mean Ribosome Loading (MRL)** predicts the MRL value for a given sequence, with target $y \in \mathbb{R}$ representing the level of mRNA translation activity into proteins. Data from Reid's dataset [89] of 91,519 5' UTR sequences and their variants are used to calculate the MRL for each sequence. The model's performance is evaluated using the $R^2$ value.

*Impact:* MRL refers to the average ribosome load on a specific mRNA sequence under given conditions, indicating the translation efficiency of ribosomes on that mRNA. Modulating the features and structures of the 5' UTR sequence can influence ribosome loading on mRNA, thereby regulating protein expression levels [62, 10].

### 3.3 Engineering Prediction

**Vaccine Degradation Prediction (VDP)** forecasts the stability and shelf life of vaccines under different environmental conditions. For each nucleotide, the three properties are recorded in target $y \in \mathbb{R}^3$. We use data from the "Stanford OpenVaccine" [111] competition on Kaggle and the RNA design platform Eterna, which includes detailed measurements for 6,043 diverse RNA constructs. The evaluation metric is the Mean Columnwise Root Mean Squared Error (MCRMSE).

*Impact:* Accurate predictions of vaccine degradation under different environmental conditions are crucial for optimizing storage and transportation protocols, ensuring vaccines remain potent until administration. Enhanced degradation predictions are particularly beneficial for distributing

vaccines in challenging environments, such as resource-limited settings, by providing guidelines to maintain vaccine stability and efficacy.

**Programmable RNA Switches (PRS)** involves identifying synthetic RNA molecules that can alter their conformation and function in response to specific signals. The target $y \in \mathbb{R}^3$ records the ON, OFF and ON/OFF states activity given an RNA sequence. The dataset, analyzed by Angenent-Mari [6] , includes 91,534 toehold switches in vivo, covering 23 viral genomes and 906 human transcription factors, with GFP signal intensity measurements indicating ON and OFF states activity levels [6]. The $R^2$ metric evaluates the effectiveness of these switches.

*Impact:* Programmable RNA switches provide precise control of gene expression and cellular functions, serving as powerful tools for investigating biological processes [99, 72]. In therapeutic applications, these switches hold promise for developing targeted and personalized treatments by responding to disease-specific signals, offering innovative approaches to medical intervention.

**CRISPR On-Target Prediction (CRI-On)** evaluates the efficiency of single-guide RNAs (sgRNAs) directed by Cas proteins in gene editing within specific target sites. Each sgRNA's knockout efficacy is quantified and presented as target $y \in \mathbb{R}$. The dataset [19] comprises approximately 15,000 sgRNAs targeting 1,071 genes across four different cell lines, with performance evaluated using the Weighted Spearman correlation coefficient.

*Impact:* CRISPR-Cas technology has transformed genetic engineering with significant enhancements in genome editing accuracy and safety. Effective on-target predictions are essential for designing sgRNAs that precisely modify genetic sequences without affecting unintended regions, thus improving therapeutic outcomes and research accuracy [116, 47].

**CRISPR Off-Target Prediction (CRI-Off)** assesses the likelihood and frequency of CRISPR-induced mutations at unintended genomic locations. The efficacy of sgRNA specificity is quantified using a target $y \in \mathbb{R}$, capturing the frequency of off-target cleavage. The evaluation dataset [19] contains data for about 160,000 potential off-target sites across 30 sgRNAs in various cell types, with the Weighted Spearman correlation coefficient serving as the metric.

*Impact:* Precision in off-target predictions is critical for advancing CRISPR technology by reducing unintended genetic modifications, which can lead to harmful effects. Accurate off-target analysis helps refine sgRNA designs, enhancing the safety and efficacy of CRISPR applications in clinical settings and research.

## 4 Models

We consider three types of baseline models in our benchmark, including naive supervised models, pre-trained language models, and the proposed BEACON-B. We give the details in the following part and summarize them in Table 2.

**Naive Supervised Models.** We utilize three widely-used sequence encoders: CNN [93], ResNet [78], and LSTM [78]. We mainly follow the design choices described in [117], employing 2 layers for CNN, 8 resblocks for ResNet, and 3 Bi-LSTM layers for LSTM, with 5.4M, 11M, and 26.7M parameters, respectively.

**Pre-trained Language Models.** We evaluate the performance of several language models, including RNA-FM [13], RNABERT [1], RNA-MSM [124], SpliceBERT [15], 3UTRBERT [119], and UTR-LM [18]. These models vary significantly in size, ranging from 0.48M to 99.52M parameters, and are pre-trained on diverse RNA data sources including ncRNA, pre-mRNA, mRNA-3'UTR, and mRNA-5'UTR. For consistency, we choose to fine-tune them using identical settings.

**Baseline RNA LM and BEACON-B .** We conduct ablation studies on two key aspects of RNA LM: 1) tokenization methods including Single Nucleotide (Single), Byte-Pair Encodings (BPE) [92, 125], Overlapping K-mer (K-mer, we use 6mer for experiments) [119, 54] and Non-overlapping K-mer (Non-overlap) [23] 2) positional encodings including Absolute Positional Encodings (APE) [28], Attention with Linear Biases (ALiBi) [76] and Rotary Positional Encodings (RoPE) [100]. The findings indicate that single nucleotide tokenization outperforms both K-mer, BPE, and Non-overlap, and ALiBi shows advantages over both RoPE and APE. Consequently, we propose a robust yet efficient BEACON baseline (BEACON-B) that incorporates single nucleotide tokenization and ALiBi as positional encodings, based on the BERT backbone.

Table 2: Detailed specifications and pre-training data of RNA language models analyzed in the study.

| RNA Foundation Model | Number of Parameters (M) | Max Token length | Pre-trained Data | Tokenizer | Positional Encoding |
|---|---|---|---|---|---|
| RNA-FM [13] | 99.52 | 1024 | Multispecies ncRNA [104] | Single | APE |
| RNABERT [1] | 0.48 | 440 | Human ncRNA [104] | Single | APE |
| RNA-MSM [124] | 95.92 | 1024 | Homologous sequences [57, 14] | Single | APE |
| SpliceBERT-H510 [15] | 19.45 | 510 | Human pre-mRNA [42] | Single | APE |
| SpliceBERT-MS510 [15] | 19.45 | 510 | Multispecies pre-mRNA [42] | Single | APE |
| SpliceBERT-MS1024 [15] | 19.72 | 1024 | Multispecies pre-mRNA [42] | Single | APE |
| UTR-LM-MRL [18] | 1.21 | 1026 | Multispecies 5'UTR [21, 88, 10] | Single | RoPE |
| UTR-LM-TE&EL [18] | 1.21 | 1026 | Multispecies 5'UTR [21, 88, 10] | Single | RoPE |
| 3UTRBERT-3mer [119] | 86.14 | 512 | Human 3'UTR [44] | K-mer | APE |
| 3UTRBERT-4mer [119] | 86.53 | 512 | Human 3'UTR [44] | K-mer | APE |
| 3UTRBERT-5mer [119] | 88.45 | 512 | Human 3'UTR [44] | K-mer | APE |
| 3UTRBERT-6mer [119] | 98.05 | 512 | Human 3'UTR [44] | K-mer | APE |

## 5 Results

### 5.1 Training Setups

To ensure a fair comparison, we fully fine-tune all the BERT-like RNA foundation models including RNA-FM, RNABERT, RNA-MSM, SpliceBERT, 3UTRBERT, UTR-LM and BEACON-B under the same training settings. For simple supervised methods (CNN, ResNet and LSTM) and baseline RNA LM, we train them from scratch using similar training settings. For each model, we search for its learning rate from 1e-5 to 5e-3. All experiments are repeated with three random seeds, and we report the average performance alongside sample standard deviations. More details are in Appendix A.1.

### 5.2 Task Pipeline

Our approach incorporates three pipelines for different types of tasks in the BEACON. In nucleotide-level tasks, due to the complexity of outputs in structural tasks, we further categorize the tasks of Secondary Structure, Contact Map, and Distance Map into a more detailed nucleotide-nucleotide level prediction.

**Sequence Level Prediction** For sequence-level tasks, we apply an attentive weighted sum of all nucleotides for naive supervised models and use the [CLS] token from language models. Both representations are processed through an MLP layer to derive the sequence-level predictions.

**Nucleotide Level Prediction** For tasks requiring resolution at the nucleotide level, individual representations for each nucleotide are processed through a Multilayer Perceptron (MLP) to generate nucleotide-level predictions. Specifically, the representation for a nucleotide are calculated by averaging the representations of all tokens that cover it, as illustrated in Appendix Fig. 2.

**Nucleotide-Nucleotide Relation Prediction** To analyze relationships between nucleotides, we compute a self outer product of the nucleotide representations to form a matrix that cotains the pairwise interactions between nucleotides. This matrix is then passed through a simple Resnet to get the final output.

### 5.3 Benchmark Results

In Table 3, we report the benchmark results for popular and opensource methods, including literature SOTAs, naive supervised models and existing RNA language models.

**ResNet and LSTM are strong naive supervised models.** ResNet, which has only been trained on downstream tasks, can outperform most if not all language models on some tasks. LSTM outperforms the other Naive supervised Models on 9 out of 13 tasks, and the performance is better by a large margin on many tasks.

**Pre-trained RNA language models have good potential for RNA understanding.** It outperforms the previous task-specific SOTA on 8 out of 13 tasks, demonstrating that the additional unsupervised pre-training brings a lot of gains. However, there is still a long way to go on individual tasks, such as contact map prediction and distance map prediction in the structural task, vaccine degradation rate prediction in the engineering task, and CRISPR on- and off-target prediction. Of course, the previous SOTA method used additional features such as secondary structure, but it shows that there is still a lot of room for improvement in the RNA language model.

Table 3: Benchmark results across various 13 RNA tasks. We use four color scales of blue to denote the first, second, third and fourth best performance among naive supervised models and pre-trained RNA LMs. Mean (std) is reported for each experiment.

| Task | SSP | CMP | DMP | SSI | SPL | APA | NcRNA | Modif | MRL | VDP | PRS | CRI-On | CRI-Off |
|---|---|---|---|---|---|---|---|---|---|---|---|---|---|
| Metric | F1 (%) | P@L (%) | $R^2$ (%) | $R^2$ (%) | ACC@K (%) | $R^2$ (%) | ACC (%) | AUC (%) | $R^2$ (%) | MCRMSE↓ | $R^2$ (%) | SC (%) | SC (%) |
| **Literature SOTA** | | | | | | | | | | | | | |
| Literature | UFold [36] | RNACon [102] | SS+Seq [13] | StructImp [40] | SpliceAI [53] | APARENT [9] | GCN [84] | MultiRM [97] | Optimus [89] | NAttn [82] | MLP-O [6] | SSC [116] | DeepCRI [19] |
| SOTA | 65.4 | 66 | 68.75 | 37.2 | $32.18_{(0.64)}$ | $50.82_{(7.00)}$ | 85.73 | 84 | 78 | 0.263 | 55.67 | 44.1 | 12.6 |
| **Naive supervised Model** | | | | | | | | | | | | | |
| CNN | $49.95_{(0.82)}$ | $43.89_{(5.53)}$ | $27.76_{(5.00)}$ | $34.36_{(0.12)}$ | $8.43_{(0.38)}$ | $50.93_{(0.17)}$ | $88.62_{(0.71)}$ | $70.87_{(0.40)}$ | $74.13_{(0.58)}$ | $0.361_{(0.003)}$ | $45.40_{(0.66)}$ | $29.69_{(2.52)}$ | $11.40_{(0.10)}$ |
| ResNet | $57.26_{(3.14)}$ | $59.59_{(0.68)}$ | $30.26_{(1.81)}$ | $37.74_{(0.16)}$ | $21.15_{(1.56)}$ | $56.45_{(0.94)}$ | $88.33_{(1.22)}$ | $71.03_{(0.32)}$ | $74.34_{(0.22)}$ | $0.349_{(0.003)}$ | $55.21_{(0.28)}$ | $28.55_{(2.42)}$ | $11.50_{(0.22)}$ |
| LSTM | $58.61_{(0.21)}$ | $40.41_{(1.67)}$ | $44.77_{(0.47)}$ | $35.44_{(1.13)}$ | $36.66_{(1.83)}$ | $67.03_{(0.86)}$ | $88.78_{(0.10)}$ | $94.83_{(0.31)}$ | $83.94_{(0.08)}$ | $0.329_{(0.002)}$ | $55.45_{(0.71)}$ | $26.83_{(1.32)}$ | $8.60_{(0.13)}$ |
| **Pretrained RNA Language Model** | | | | | | | | | | | | | |
| RNA-FM | $68.50_{(0.54)}$ | $47.56_{(6.73)}$ | $51.45_{(0.51)}$ | $42.36_{(0.24)}$ | $34.84_{(0.87)}$ | $70.32_{(0.97)}$ | $96.81_{(0.061)}$ | $94.98_{(0.042)}$ | $79.47_{(0.47)}$ | $0.347_{(0.05)}$ | $55.98_{(0.09)}$ | $31.62_{(1.16)}$ | $2.49_{(1.56)}$ |
| RNABERT | $57.27_{(0.30)}$ | $45.21_{(10.87)}$ | $48.19_{(0.64)}$ | $31.62_{(0.64)}$ | $0.18_{(0.18)}$ | $57.66_{(2.11)}$ | $68.95_{(7.285)}$ | $82.82_{(19.09)}$ | $29.79_{(20.15)}$ | $0.378_{(0.003)}$ | $54.60_{(0.23)}$ | $29.77_{(3.98)}$ | $4.27_{(1.05)}$ |
| RNA-MSM | $57.98_{(0.47)}$ | $57.26_{(15.38)}$ | $37.49_{(4.10)}$ | $39.22_{(0.23)}$ | $38.33_{(0.76)}$ | $70.40_{(1.12)}$ | $84.85_{(0.266)}$ | $94.89_{(0.14)}$ | $83.48_{(0.18)}$ | $0.330_{(0.001)}$ | $56.94_{(0.38)}$ | $34.92_{(1.99)}$ | $3.85_{(0.99)}$ |
| Splice-H510 | $64.93_{(0.84)}$ | $45.80_{(6.03)}$ | $55.56_{(1.00)}$ | $38.91_{(0.07)}$ | $44.80_{(1.93)}$ | $58.65_{(2.34)}$ | $95.92_{(0.666)}$ | $62.57_{(1.92)}$ | $83.49_{(0.47)}$ | $0.321_{(0.000)}$ | $54.90_{(3.45)}$ | $26.61_{(1.30)}$ | $4.00_{(1.13)}$ |
| Splice-MS510 | $43.24_{(28.64)}$ | $52.64_{(7.56)}$ | $10.27_{(0.20)}$ | $38.58_{(0.50)}$ | $50.55_{(0.49)}$ | $52.46_{(17.36)}$ | $95.87_{(0.364)}$ | $96.05_{(0.777)}$ | $84.89_{(0.28)}$ | $0.315_{(0.003)}$ | $50.98_{(7.46)}$ | $27.13_{(0.27)}$ | $3.49_{(2.12)}$ |
| Splice-MS1024 | $68.26_{(0.20)}$ | $47.32_{(3.16)}$ | $55.89_{(0.48)}$ | $39.22_{(0.02)}$ | $48.52_{(0.49)}$ | $60.03_{(3.42)}$ | $96.05_{(0.777)}$ | $53.45_{(6.25)}$ | $67.15_{(30.54)}$ | $0.313_{(0.000)}$ | $57.72_{(0.45)}$ | $27.59_{(4.61)}$ | $5.00_{(0.71)}$ |
| UTR-LM-MRL | $59.71_{(0.30)}$ | $45.51_{(23.51)}$ | $55.21_{(2.91)}$ | $39.52_{(0.36)}$ | $36.20_{(1.84)}$ | $64.99_{(4.90)}$ | $89.97_{(0.617)}$ | $56.41_{(2.90)}$ | $77.78_{(6.03)}$ | $0.325_{(0.002)}$ | $57.28_{(0.10)}$ | $28.49_{(1.37)}$ | $4.28_{(0.15)}$ |
| UTR-LM-TE&EL | $59.57_{(0.20)}$ | $60.32_{(7.27)}$ | $54.94_{(2.54)}$ | $40.15_{(0.11)}$ | $37.35_{(5.48)}$ | $72.09_{(0.82)}$ | $81.33_{(8.551)}$ | $59.70_{(10.52)}$ | $82.50_{(1.45)}$ | $0.319_{(0.001)}$ | $53.37_{(3.54)}$ | $32.49_{(4.14)}$ | $2.91_{(1.18)}$ |
| UTRBERT-3mer | $60.37_{(0.47)}$ | $51.03_{(21.48)}$ | $50.95_{(0.44)}$ | $44.31_{(0.00)}$ | $44.24_{(0.53)}$ | $69.52_{(4.56)}$ | $92.88_{(0.379)}$ | $95.14_{(0.11)}$ | $83.89_{(0.13)}$ | $0.320_{(0.001)}$ | $56.83_{(0.26)}$ | $29.92_{(1.95)}$ | $4.48_{(1.12)}$ |
| UTRBERT-4mer | $59.41_{(0.45)}$ | $44.91_{(27.56)}$ | $47.77_{(2.08)}$ | $33.22_{(0.00)}$ | $42.04_{(0.53)}$ | $72.71_{(0.85)}$ | $94.32_{(0.946)}$ | $95.10_{(0.12)}$ | $82.90_{(0.75)}$ | $0.341_{(0.000)}$ | $56.43_{(0.67)}$ | $23.20_{(0.11)}$ | $3.11_{(1.10)}$ |
| UTRBERT-5mer | $47.92_{(8.75)}$ | $44.71_{(7.64)}$ | $48.67_{(1.70)}$ | $31.27_{(0.00)}$ | $39.19_{(0.37)}$ | $72.70_{(1.77)}$ | $93.04_{(0.367)}$ | $94.78_{(0.07)}$ | $75.64_{(4.70)}$ | $0.343_{(0.001)}$ | $57.16_{(0.08)}$ | $25.74_{(0.00)}$ | $3.93_{(0.24)}$ |
| UTRBERT-6mer | $38.56_{(28.76)}$ | $51.56_{(20.30)}$ | $50.02_{(1.05)}$ | $29.93_{(0.17)}$ | $38.58_{(2.72)}$ | $71.17_{(2.30)}$ | $93.12_{(0.168)}$ | $95.08_{(0.17)}$ | $83.60_{(0.39)}$ | $0.340_{(0.001)}$ | $57.14_{(0.12)}$ | $28.60_{(1.55)}$ | $4.90_{(0.57)}$ |
| **Our BEACON-B** | | | | | | | | | | | | | |
| BEACON-B | $64.18_{(0.44)}$ | $60.81_{(1.70)}$ | $56.28_{(0.41)}$ | $38.78_{(0.18)}$ | $37.43_{(1.43)}$ | $70.59_{(0.91)}$ | $94.63_{(0.16)}$ | $94.74_{(0.20)}$ | $72.29_{(0.28)}$ | $0.320_{(0.001)}$ | $54.67_{(0.36)}$ | $26.01_{(1.81)}$ | $4.42_{(0.33)}$ |
| BEACON-B512 | $58.75_{(3.72)}$ | $61.20_{(2.11)}$ | $56.82_{(0.63)}$ | $39.13_{(0.08)}$ | $37.24_{(1.09)}$ | $72.00_{(0.17)}$ | $94.99_{(0.21)}$ | $94.92_{(0.07)}$ | $72.35_{(0.28)}$ | $0.320_{(0.001)}$ | $55.20_{(0.26)}$ | $28.17_{(1.81)}$ | $3.82_{(1.04)}$ |

**SpliceBERT and RNA-FM are superior models for various tasks.** Both SpliceBERT-MS1024 and RNA-FM got first place in 3 out of 13 tasks, and had top performances in other tasks as well, showing they have learned rich patterns and evolution knowledge from multi-species RNA sequences.

**Pre-training of specific RNA attributes will result in a gain on tasks with corresponding attributes.** First, when specific RNA attributes are included in the pre-training it brings gains to the downstream tasks corresponding to the attributes. For example, RNA-FM pre-trained with non-coding RNA achieves the best performance in non-coding RNA family prediction, SpliceBERT pre-trained on pre-mRNA learns information about potential shear mRNAs for the best shear site prediction, and 3UTRBERT uses sequences from the 3'UTR region to learn 3'UTR function worked best in the prediction of APA isoforms in the 3'UTR functional region, and similarly, the pre-trained UTR-LM in the 5'UTR region worked well in the prediction of ribosome loading in the 5'UTR association. Second, specific attributes also give gains for having other RNA attributes, for example, 3UTRBERT, although pre-trained on 3'UTR sequences, also gained on the prediction of 5'UTR function.

Table 4: Performance of baseline RNA LMs with different tokenizers and positional encodings.

| Task | SSP | CMP | DMP | SSI | SPL | APA | NcRNA | Modif | MRL | VDP | PRS | CRI-On | CRI-Off |
|---|---|---|---|---|---|---|---|---|---|---|---|---|---|
| Metric | F1 (%) | P@L (%) | $R^2$ (%) | $R^2$ (%) | ACC@K (%) | $R^2$ (%) | ACC (%) | AUC (%) | $R^2$ (%) | MCRMSE↓ | $R^2$ (%) | SC (%) | SC (%) |
| **Baseline RNA LM Analysis** | | | | | | | | | | | | | |
| Non-overlap-APE | $12.58_{(0.08)}$ | $41.60_{(5.34)}$ | $44.66_{(0.59)}$ | $10.36_{(0.57)}$ | $0.00_{(0.00)}$ | $58.49_{(0.75)}$ | $82.04_{(0.18)}$ | $79.07_{(20.82)}$ | $34.56_{(2.79)}$ | $0.640_{(0.000)}$ | $51.95_{(0.15)}$ | $21.64_{(3.85)}$ | $8.33_{(0.90)}$ |
| Non-overlap-ALiBi | $5.58_{(0.49)}$ | $57.49_{(32.45)}$ | $37.44_{(2.69)}$ | $10.84_{(1.13)}$ | $0.00_{(0.00)}$ | $57.92_{(0.36)}$ | $80.55_{(1.38)}$ | $60.56_{(9.00)}$ | $37.75_{(1.07)}$ | $0.640_{(0.000)}$ | $49.17_{(0.17)}$ | $14.85_{(4.09)}$ | $7.95_{(0.20)}$ |
| Non-overlap-RoPE | $4.50_{(0.16)}$ | $27.47_{(22.17)}$ | $38.68_{(0.39)}$ | $10.38_{(0.94)}$ | $0.00_{(0.00)}$ | $46.14_{(0.62)}$ | $76.76_{(0.44)}$ | $24.13_{(18.27)}$ | | $0.640_{(0.000)}$ | $31.76_{(0.04)}$ | $15.76_{(0.27)}$ | $8.43_{(0.22)}$ |
| BPE-APE | $6.30_{(0.51)}$ | $49.48_{(3.75)}$ | $41.56_{(0.16)}$ | $20.79_{(0.95)}$ | $0.00_{(0.00)}$ | $65.75_{(1.16)}$ | $80.76_{(0.95)}$ | $67.67_{(0.28)}$ | $48.67_{(1.45)}$ | $0.641_{(0.001)}$ | $29.75_{(0.08)}$ | $16.16_{(2.77)}$ | $5.78_{(1.50)}$ |
| BPE-ALiBi | $6.34_{(0.85)}$ | $59.17_{(18.04)}$ | $37.58_{(1.10)}$ | $25.23_{(1.67)}$ | $0.00_{(0.00)}$ | $69.03_{(1.05)}$ | $81.36_{(0.43)}$ | $63.95_{(4.77)}$ | $45.78_{(5.49)}$ | $0.642_{(0.001)}$ | $31.45_{(0.79)}$ | $16.13_{(3.84)}$ | $6.40_{(1.44)}$ |
| BPE-RoPE | $6.32_{(0.38)}$ | $51.31_{(23.54)}$ | $37.01_{(0.47)}$ | $21.96_{(1.13)}$ | $0.00_{(0.00)}$ | $50.91_{(1.27)}$ | $75.74_{(1.10)}$ | $62.89_{(4.01)}$ | $45.87_{(1.53)}$ | $0.642_{(0.001)}$ | $19.69_{(0.10)}$ | $16.46_{(1.75)}$ | $6.63_{(0.73)}$ |
| Single-APE | $48.23_{(0.26)}$ | $70.85_{(8.03)}$ | $48.22_{(0.69)}$ | $24.38_{(12.80)}$ | $0.18_{(0.00)}$ | $56.35_{(2.65)}$ | $84.81_{(0.74)}$ | $93.51_{(0.24)}$ | $1.45_{(0.22\downarrow)}$ | $0.376_{(0.003)}$ | $55.22_{(0.38)}$ | $35.51_{(0.63)}$ | $5.66_{(0.20)}$ |
| Single-ALiBi | $49.78_{(0.34)}$ | $49.70_{(1.05)}$ | $42.62_{(6.15)}$ | $38.84_{(1.01)}$ | $28.27_{(0.91)}$ | $66.15_{(2.92)}$ | $88.18_{(0.57)}$ | $73.62_{(14.68)}$ | $69.04_{(5.30)}$ | $0.347_{(0.002)}$ | $51.68_{(0.50)}$ | $22.27_{(0.22)}$ | $5.66_{(0.47)}$ |
| Single-RoPE | $39.20_{(0.82)}$ | $51.64_{(0.38)}$ | $15.72_{(3.01)}$ | $10.15_{(0.07)}$ | $0.00_{(0.00)}$ | $33.34_{(1.17)}$ | $38.71_{(2.50)}$ | $65.05_{(0.64)}$ | $1.36_{(0.22\downarrow)}$ | $0.462_{(0.000)}$ | $14.59_{(0.20)}$ | $21.11_{(0.08)}$ | $4.89_{(0.09)}$ |
| 6mer-APE | $16.24_{(0.54)}$ | $55.65_{(23.35)}$ | $43.21_{(0.83)}$ | $12.24_{(0.90)}$ | $13.92_{(0.86)}$ | $52.59_{(6.86)}$ | $87.63_{(0.91)}$ | $91.71_{(0.93)}$ | $67.75_{(0.89)}$ | $0.420_{(0.009)}$ | $51.39_{(0.46)}$ | $9.99_{(2.58)}$ | $4.21_{(1.08)}$ |
| 6mer-ALiBi | $13.99_{(0.35)}$ | $28.45_{(9.44)}$ | $39.48_{(2.99)}$ | $11.49_{(0.39)}$ | $22.82_{(0.83)}$ | $58.93_{(0.09)}$ | $87.49_{(0.86)}$ | $60.82_{(2.72)}$ | $69.01_{(1.04)}$ | $0.417_{(0.001)}$ | $48.26_{(0.44)}$ | $9.70_{(4.31)}$ | $4.64_{(0.88)}$ |
| 6mer-RoPE | $22.18_{(0.90)}$ | $37.95_{(9.18)}$ | $36.93_{(1.02)}$ | $12.89_{(0.41)}$ | $7.46_{(0.65)}$ | $45.71_{(0.30)}$ | $86.55_{(0.62)}$ | $64.41_{(5.70)}$ | $66.38_{(0.90)}$ | $0.435_{(0.005)}$ | $35.17_{(0.46)}$ | $9.91_{(2.12)}$ | $5.53_{(1.03)}$ |

## 5.4 Component Analysis of RNA Language Models

In Table 4 , we study the language model component effect from tokenizer and positional encoding.

**The single nucleotide tokenizer is a powerful tool for RNA language models.** As shown in Table 17, it achieves the best performance on 11 out of 13 tasks, significantly outperforming other tokenizers. BPE and non-overlapping tokenizers are generally ineffective at the nucleotide level, as they lose precision from overlapping. Similarly, the 6mer approach adds local information before the individual tokens, potentially introducing redundancy. We argue that the single nucleotide tokenizer can learn global information, including surrounding context, through self-attention mechanisms.

Thus, using the single nucleotide tokenizer is sufficient, and future work should focus on designing interpretable tokenizers based on single nucleotide units [107, 120, 66].

**ALiBi is better for RNA sequences understanding.** For tasks involving shorter sequences, the specific advantages of RoPE or other complex encoding schemes may not be fully realized. RoPE, which is highly effective in long sequences due to its rotational component that maintains relative positioning across long distances, might not provide significant benefits over simpler methods like ALiBi in shorter sequences. Moreover, ALiBi linearly biases the attention scores based on relative positions, which helps the model better generalize across different sequence lengths.

Table 5: Comparison of GPU days and the number of tasks (total: 13) where BEACON-B demonstrates performance superiority over other methods.

| Model | BEACON -B | BEACON -B512 | RNA-FM | SpliceBERT -H510 | SpliceBERT -MS510 | SpliceBERT -MS1024 | UTRBERT -3mer | UTRBERT -4mer | UTRBERT -5mer | UTRBERT -6mer |
|---|---|---|---|---|---|---|---|---|---|---|
| # GPU Days | 1.3*8=**10.4** | 0.895*4=**3.58** | 30*8=240 | 7*8=56 | 7*8=56 | 7*8+3*4=68 | 38*4=152 | 38*4=152 | 38*4=152 | 38*4=152 |
| # Tasks where BEACON-B Performs Better | - | - | 6 | 6 | 8 | 5 | 7 | 8 | 8 | 6 |
| # Tasks where BEACON-B512 Performs Better | - | - | 6 | 8 | 9 | 6 | 6 | 7 | 8 | 7 |

## 5.5 BEACON-B: an Efficient Baseline for RNA Language Models

Based on the above analysis of the different components of the RNA language model, combined with the Table 4, we use the single nucleotide tokenizer, ALiBi as the positional encoding, and pre-train on filtered human ncRNA sequences from RNACentral [104]. We propose the low-resource and cost-effective BEACON-B (pre-trained on 1026 length seqs) and BEACON-B512 (pre-trained on 512 length seqs and FlashAttn [26]) as a baseline to provide an extremely fast and easy to use open-source pre-training model for subsequent researchers.

Although with very small GPU days (days * GPUs) as the cost of pre-training, BEACON-B can even outperform SOTA pre-trained RNA LMs on some tasks such as **contact map prediction** and **distance map prediction**. Compared with other models that also report pre-training resources, BEACON-B and BEACON-B512 can match or even surpass existing RNA language models in one-to-one comparisons on **almost half of the tasks** listed in Table 3 and Table 5, despite being pre-trained with significantly fewer resources. This demonstrates that the insights we obtain from the important components in analysing the RNA language model are vital and that biological motifs and configurations on limited RNA data can be fully explored by utilising such a combination of components in a good way.

## 6 Conclusions

**Summary.** In this work, we present BEACON, the first comprehensive RNA benchmark, which encompasses 13 diverse tasks spanning structural analysis, functional studies, and engineering applications. BEACON aims to address the critical gap in standardized evaluation for RNA models. We assess various models, from traditional approaches like CNNs to advanced RNA foundation models, providing insights into their task-specific performances. Additionally, we analysis the vital components of RNA LM from tokenization and positional encoding. Building upon this, we propose BEACON-B , an efficient baseline that incorporates single nucleotide tokenization and ALiBi. BEACON's standardized evaluation framework and the insights provided into RNA modeling components are expected to significantly advance RNA research, facilitating the development of more sophisticated models and enhancing our understanding of RNA's diverse roles in biology.

**Limitation & future work.** Despite the comprehensiveness of BEACON, it has some limitations for future work. While BEACON includes 13 diverse RNA-related tasks, it may not cover all aspects of RNA biology, necessitating the inclusion of additional tasks and datasets in future versions. The influence of pre-training datasets and hyperparameters on model performance also needs further systematic exploration to optimize configurations for specific RNA tasks. Although BEACON-B serves as an efficient baseline, there is potential for developing more advanced models that leverage RNA's unique structural characteristics. Additionally, BEACON primarily evaluates predictive accuracy, suggesting the need to incorporate metrics like interpretability, computational efficiency, and robustness for a more holistic assessment. Addressing these limitations and exploring new directions will not only advance RNA research but also deepen our understanding of its indispensable roles in genetic regulation, disease pathogenesis, and therapeutic development.

## Acknowledgments

This work is funded in part by Shanghai Artificial Intelligence Laboratory and supported in part by HKU Startup Fund, HKU Seed Fund for Basic Research, HKU Seed Fund for Translational and Applied Research, HKU IDS research Seed Fund, and HKU Fintech Academy R&D Funding. It is also supported by the Beijing Super Cloud Computing Center serve platform.

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

# A Appendix

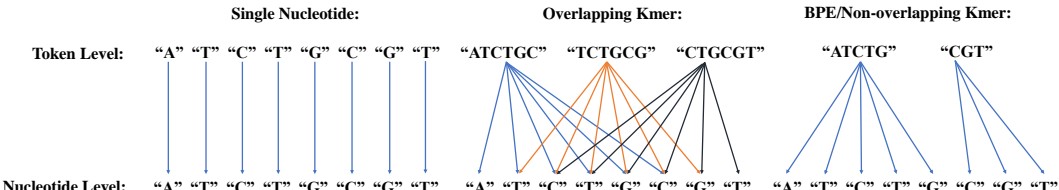

Figure 2: Derivation of nucleotide-level representations. In single nucleotide tokenization, a token directly corresponds to a nucleotide, thus the representations are identical. For overlapping Kmer tokenization, the nucleotide representation is the averaged representation of tokens covering it. In Byte-Pair Encoding (BPE) and non-overlapping K-mer tokenization, the representation is derived from the token covering it.

## A.1 Experimental Settings for Tasks

### A.1.1 Most of the Tasks

Most of the tasks are trained using the same training settings shown in the Tab 6. These tasks include structural score imputation, splice site prediction, APA isoform prediction, non-coding RNA function classification, modification prediction, programmable RNA switches, CRISPR on-target prediction and CRISPR off-target prediction.

Table 6: Configuration settings for most of the tasks training.

| config | value |
|---|---|
| optimizer | AdamW |
| optimizer epsilon | 1e-8 |
| optimizer momentum | $\beta_1, \beta_2 = 0.9, 0.999$ |
| weight decay | 0.01 |
| learning rate sch. | linear decay |
| learning rate | [1e-5,5e-3] |
| warmup steps | 50 |
| epochs | 30 |
| batch size | 32 |
| gradient accumulation | 1 |
| dtype | float16 |

In particular, for the structural score imputation task, the input is the sequence accompanied by structural scores. The sequence is fed to the model and undergoes the transformation to the nucleotide-level representation. We concatenates it with the MLP-passed structural scores, and then use the regression header to get imputation scores.

For CRISPR off-target, the input is two sequences including sgRNA and target sequences. We feed them through the same model separately, then concat them together. Finally, we use the regression header to get the predicted value of off-target.

### A.1.2 Vaccine Degradation Prediction Task

Vaccine degradation prediction is trained using the settings shown in the Table 7.

### A.1.3 Nucleotide-Nucleotide Level Tasks

Nucleotide-nucleotide level tasks are trained using the settings shown in the Table 8. In addition, the representation also follows Fig 2.

Table 7: Configuration settings for vaccine degradation prediction training.

| config | value |
| --- | --- |
| optimizer | AdamW |
| optimizer epsilon | 1e-8 |
| optimizer momentum | $\beta_1, \beta_2 = 0.9, 0.999$ |
| weight decay | 0.01 |
| learning rate sch. | linear decay |
| learning rate | [1e-5,5e-3] |
| warmup steps | 50 |
| epochs | 100 |
| batch size | 32 |
| gradient accumulation | 1 |
| dtype | float16 |

Table 8: Configuration settings for nucleotide-nucleotide level tasks training.

| config | value |
| --- | --- |
| optimizer | Adam |
| optimizer epsilon | 1e-8 |
| optimizer momentum | $\beta_1, \beta_2 = 0.9, 0.999$ |
| learning rate sch. | cosine decay |
| learning rate | [1e-5,5e-3] |
| warmup epochs | 1 |
| epochs | 100 |
| batch size | 1 |
| gradient accumulation | 8 |
| dtype | float16 |

## A.2 Detailed Data Preprocessing and More Baselines for Each Task

### A.2.1 Secondary Structure Prediction

We followed the preprocessing from the bpRNA-1m dataset [95]. To mitigate sequence redundancy and enhance the diversity of our dataset, we applied an 80% sequence-identity cut-off and restricted the maximum sequence length to below 500 nucleotides, similar to the process described in both referenced articles. This step was crucial in reducing the risk of overfitting and ensuring that our models are trained on genetically distinct samples.

The dataset was strategically split into three parts: a training set (TR0), a validation set (VL0), and a test set (TS0). This separation was carried out randomly to avoid any biases that might affect the evaluation of the model's performance.

More baselines on SSP are shown in Table A.2.1.

### A.2.2 Contact Map Prediction

We choose the benchmark datasets used by [103, 12], which are constructed based on a set of nonredundant RNA 3D structures, containing 1786 entries with resolution < 4°A initially. Sequences with length < 32nt or > 1000nt or with redundancy over 80% as well as with too few positive points (< 5) are removed. Finally, 221 sequences left are used for training and 80 sequences for testing. Using their pdb files, We then compute the ground truth pairwise tertiary distance, which is defined as the minimal atomic distance of the two bases. Then the pairwise contact is defined as the distance between two bases is less than 8°A.

More baselines on CMP are shown in Table A.2.2.

Table 9: Comparison on SSP.

| Method | Metric (F1%) |
|---|---|
| **Literature SOTA** | |
| Contextfold [123] | 54.6 |
| CONTRAfold [29] | 56.7 |
| E2Efold [17] | 13.0 |
| RNAstructure [70] | 53.3 |
| RNAsoft [5] | 53.5 |
| RNAfold [67] | 53.6 |
| Mfold [127] | 53.8 |
| LinearFold [50] | 55.0 |
| MXfold2 [91] | 55.8 |
| Externafold [110] | 56.3 |
| SPOT-RNA [95] | 61.9 |
| UFold | **65.4** |
| **Naive Supervised Model** | |
| CNN | 49.95±0.82 |
| ResNet | 57.26±3.14 |
| LSTM | **58.61±0.21** |
| **Pretrained RNA Language Model** | |
| RNA-FM | **68.50±0.54** |
| RNABERT | 57.27±0.30 |
| RNA-MSM | 57.98±0.47 |
| SpliceBERT-H510 | 64.93±0.84 |
| SpliceBERT-MS510 | 43.24±28.64 |
| SpliceBERT-MS1024 | 68.26±0.20 |
| UTR-LM-MRL | 59.71±0.30 |
| UTR-LM-TE&EL | 59.57±0.20 |
| UTRBERT-3mer | 60.37±0.47 |
| UTRBERT-4mer | 59.41±0.45 |
| UTRBERT-5mer | 47.92±8.75 |
| UTRBERT-6mer | 38.56±28.76 |
| **Our BEACON-B** | |
| BEACON-B | **64.18±0.44** |
| BEACON-B512 | 58.75±3.72 |

### A.2.3 Distance Map Prediction

The dataset used for distance map prediction is the same as the contact map prediction. We first compute the pairwise tertiary distance as above. Then we limit the distance from 0 to 20°A and regard the value over 20 as 20. Finally, we use 20 to normalize the distance values and obtain a normalized distance map with elements falling into [0, 1].

### A.2.4 Structural Score Imputation

We followed the preprocessing procedure as outlined in [40]. The icSHAPE HEK293 cell line raw sequencing data ($\sim$ 300 million clean reads) was downloaded and processed using icSHAPE-pipe to obtain structure profiles for 4,091 transcripts. The full-length transcript structure profiles were then binned into non-overlapping 100-nt fragments. 21,859 fragments were retained with valid structural scores for every nucleotide.

These valid fragments were randomly split into training and test sets in a 7:3 ratio, resulting in 15,085 fragments for training and 6,774 for testing. 569 fragments, with high sequence similarity to any fragment in the training set based on BLAST results, were removed from the test set.

For the training set, 30% of the nucleotides were randomly masked as null, with the ground-truth structural scores of these positions used as training labels. For the test set, the original icSHAPE dataset was downsampled to 100 million clean reads, with the structure profiles to simulate lower

Table 10: Comparison on CMP.

| Method | Metric (P@L %) |
|---|---|
| **Literature SOTA** | |
| PLMC [112] | 28 |
| RNAcontact (Seq) [103] | 33 |
| RNAcontact (Cov) [103] | 59 |
| RNAcontact (SS) [103] | 58 |
| RNAcontact | **66** |
| **Naive Supervised Model** | |
| CNN | 43.89±5.53 |
| ResNet | **59.59±0.68** |
| LSTM | 40.41±1.67 |
| **Pretrained RNA Language Model** | |
| RNA-FM | 47.56±6.73 |
| RNABERT | 45.21±10.87 |
| RNA-MSM | 57.26±15.38 |
| SpliceBERT-H510 | 45.80±6.03 |
| SpliceBERT-MS510 | 52.64±7.56 |
| SpliceBERT-MS1024 | 47.32±3.16 |
| UTR-LM-MRL | 45.51±23.51 |
| UTR-LM-TE&EL | **60.32±7.27** |
| UTRBERT-3mer | 51.03±21.48 |
| UTRBERT-4mer | 44.91±27.56 |
| UTRBERT-5mer | 44.71±7.64 |
| UTRBERT-6mer | 51.56±20.30 |
| **Our BEACON-B** | |
| BEACON-B | 60.81±1.70 |
| BEACON-B512 | **61.20±2.11** |

sequencing depth recalculated. This process generated missing values, and only fragments with at least one valid structural score were retained, resulting in 3,095 fragments in the final test dataset.

### A.2.5 Splice Site Prediction

We followed the preprocessing protocol from [53]. The GENCODE V24lift37 gene annotation table was downloaded from the UCSC table browser, and 20,287 protein-coding gene annotations were extracted. For genes with multiple isoforms, the principal transcript was selected. Genes lacking splice junctions were removed. The remaining genes were split into training and test sets: chromosomes 2, 4, 6, 8, 10-22, X, and Y were used for training, with 1/9 reserved for early stopping. For testing, genes from chromosomes 1, 3, 5, 7, and 9 without paralogs were selected.

For each gene, mRNA transcript sequences from the canonical transcription start to end sites were extracted using the hg19/GRCh37 assembly. Sequences were zero-padded to a multiple of 5,000 nucleotides and split into blocks of length 5,000. Then, the center 100 nucleotides of each block were extracted as the input sequence, and the corresponding splice site label was extracted as the output. (non-splice sites [1, 0, 0], splice acceptors [0, 1, 0], and splice donors [0, 0, 1])

### A.2.6 APA Isoform Prediction

The preprocessing for IPA Isoform began with filtering the raw sequencing reads from all MPRAs [94] to ensure high-quality, full-length RNA reads. These reads were clustered based on the randomized region upstream of the proximal polyadenylation site (pPAS), creating a dictionary of sequence variants for each library. To expand the dictionary, the plasmid library was sequenced to include members that did not express a distal isoform, and RNA reads were mapped to their respective dictionary entries by matching the upstream region with the shortest Hamming distance.

For each mapped read, the polyadenylation cleavage site was identified by scanning for the Poly-A tail. The cleavage positions were stored as vectors associated with each sequence variant, including a special position for reads mapping to non-random distal sites. The resulting dataset consisted of a

dictionary of unique sequence variants with associated cleavage-position count vectors, which then underwent a final filtering step. This step selected high-confidence variants by removing sequences supported by fewer than 10-20 unique UMI RNA reads or sequences with over 75% A-nucleotides in a 12-20 bp region to avoid internal priming artifacts.

Data from 12 random 3' UTR libraries were processed, with 9 used for training and 3 held out (the 3 held-out libraries were not used in the current analysis). To ensure balanced representation in the test set, the sequences from each library were individually shuffled based on the read count, then combined using a round-robin order, placing one sequence from each library after another in descending order of read count. This process ensured that the test set contained an equal number of high-read count sequences from each library. The remaining sequences were placed at the beginning of the combined library, and the training set was further shuffled. This approach ensured a balanced, high-quality dataset for training, validation, and testing. To maintain balanced and high-quality data for benchmarking, the top 10% of high-read count sequences were selected, and the most highly expressed sequences were further chosen for testing.

### A.2.7 Non-coding RNA Function Classification

We followed nRC [34] and sourced non-coding RNA sequences from the Rfam database, recognized for its comprehensive collection of manually curated RNA sequences. We specifically selected 13 diverse ncRNA classes including miRNA, 5S rRNA, 5.8S rRNA, ribozymes, CD-box, HACA-box, scaRNA, tRNA, Intron gpI, Intron gpII, IRES, leader and riboswitch. Utilizing the CD-HIT tool [35], we reduced sequence redundancy to 20%, ensuring a representative yet manageable dataset size. This process allowed us to include a wide array of RNA types while controlling data complexity.

For each selected class, we systematically assembled a training-validation dataset comprising 500 sequences per class, except for the IRES class where only 320 sequences were available, leading to a total of 6320 ncRNA sequences. We subsequently divided it into two groups: a validation set comprising 650 sequences, with 50 from each class, and a training set consisting of the remaining 5670 sequences. The balancing was further refined in our test dataset, which contains an equal number of sequences from each class, totaling 2600 sequences, ensuring that our model evaluation would not be biased by uneven class representation.

More baselines on ncRNA are shown in Table A.2.7.

### A.2.8 Modification Prediction

We followed MultiRM [97] and compiled a comprehensive dataset comprising 20 epi-transcriptome profiles derived from 15 different base-resolution technologies, addressing 12 distinct RNA modifications such as m6A, m1A, m5C, among others. A key focus was given to constructing a reliable set of negative control data, which was crucial for our predictors' accuracy. To achieve this, negative sites (non-modified nucleotides) were carefully selected from the unmodified bases within the same transcripts that contained the positive modification sites, ensuring an authentic comparison baseline.

Moreover, the dataset was divided into three distinct subsets: training, validation, and testing. The training set was purposefully left unbalanced to mirror the natural prevalence differences among the RNA modification types, which introduces realistic challenges in model training akin to real-world scenarios. Conversely, the validation and test sets were meticulously balanced—each set containing 150 and 50 samples, respectively, across all modification types—to ensure fairness in model evaluation and performance metrics.

### A.2.9 Mean Ribosome Loading

For 5'UTR processing, we primarily employed a large-scale synthetic Human 5'UTR library [88], which comprised 83,919 sequences of varying lengths, as articulated in the first referenced article. The validation set was meticulously crafted by evenly sampling 7600 sequences across these different lengths to ensure fair representation and robust generalizability testing. The remaining sequences were utilized for training purposes. To further enhance our model's performance assessment, we incorporated an additional dataset of 7600 real human 5'UTRs, mirroring the length distribution provided by the synthetic library. This dual-dataset strategy not only allowed for a thorough evaluation of the model's predictive accuracy but also its ability to generalize across synthetic and real-world data.

Table 11: Comparison on ncRNA.

| Method | Metric (ACC %) |
|---|---|
| **Literature SOTA** | |
| EDeN [71] | 67 |
| nRC [34] | 81.81 |
| RNAGCN | **85.73** |
| **Naive Supervised Model** | |
| CNN | 88.62±0.71 |
| ResNet | 88.33±1.22 |
| LSTM | **88.78±0.10** |
| **Pretrained RNA Language Model** | |
| RNA-FM | **96.81±0.061** |
| RNABERT | 68.95±7.285 |
| RNA-MSM | 84.85±0.266 |
| SpliceBERT-H510 | 95.92±0.666 |
| SpliceBERT-MS510 | 95.87±0.364 |
| SpliceBERT-MS1024 | 96.05±0.777 |
| UTR-LM-MRL | 89.97±0.617 |
| UTR-LM-TE&EL | 81.33±8.551 |
| UTRBERT-3mer | 92.88±0.379 |
| UTRBERT-4mer | 94.32±0.946 |
| UTRBERT-5mer | 93.04±0.367 |
| UTRBERT-6mer | 93.12±0.168 |
| **Our BEACON-B** | |
| BEACON-B | 94.63±0.16 |
| BEACON-B512 | **94.99±0.21** |

Additionally, as described in the second referenced article, the pivotal role of the 5'UTR sequence in translation efficiency was explored through rigorous experiments utilizing data from Massively Parallel Reporter Assays (MPRAs), which included a substantial library of 280,000 gene sequences.

### A.2.10 Vaccine Degradation Prediction

We followed OpenVaccine [111] to collect 2400 sequences specifically for training purposes, 629 sequences were made available for public testing during the competition, and the remainder were reserved for private scoring. Post-competition, all these sequences are now available with comprehensive labels detailing various degradation rates under multiple experimental conditions, such as high pH and high temperature, both with and without magnesium.

To ensure the integrity of the public leaderboard and prevent biases, we implemented rigorous filtering criteria on the 629 sequences used for public testing. These criteria included setting a minimum value threshold across all experimental conditions and ensuring a mean signal-to-noise ratio above a specified level.

We also provide the test data of the original private leaderboard although we do not use it in the paper. For the private leaderboard data, which involves the most critical evaluation, we included measurements from an additional 3005 RNA sequences, which are slightly longer. These were processed to include data for the first 91 bases, carefully excluding the final bases to align with experimental limitations.

### A.2.11 Programmable RNA Switches

We followed the data generation pipeline from [7]. A toehold-switch library based on 244,000 putative trigger sequences was designed and synthesized, which covered the complete genomes of 23 pathogenic viruses, the entire coding regions of 906 human transcription factors, and approximately 10,000 random sequences. The synthesized oligo pool was used to generate two construct libraries for ON and OFF states, which were transformed into BL21 E. coli. The OFF library contained toehold-switch constructs without triggers, while the ON library contained the same toeholds with complementary triggers fused to their corresponding switches.

These libraries were sorted using fluorescence-activated cell sorting (FACS) into four bins, and the variants in each bin were quantified using next-generation sequencing (NGS) to determine their fluorescence distributions. After quality control, the toehold-switch library included 109,067 ON-state measurements, 163,967 OFF-state measurements, and 91,534 ON/OFF paired ratios, where both states were characterized for a given switch. The ON and OFF data were normalized from 0 to 1, resulting in ON/OFF ratios normalized from -1 to 1. Following [7], a quality control process was applied to remove artifacts and ensure the reliability of the data. There are 5 levels of quality control (QC1, QC2, QC3, QC4, and QC5), with QC1 being the lowest quality and QC5 being the highest. All datasets at QC levels above QC2 are used for training with Q5 left for testing.

Table 12: Comparison on CRI-On.

| Method | Metric (SC %) |
|---|---|
| **Literature SOTA** | |
| CHOPCHOP [60] | 0.9 |
| DeepCRISPR [19] | 26.2 |
| WU-CRISPR [114] | 26.8 |
| sgRNA Scorer [11] | 30.5 |
| CRISPR MultiTargeter [77] | 32.5 |
| E-CRISP [45] | 33.6 |
| sgRNA Designer [30] | 41.8 |
| SSC | **44.1** |
| **Naive Supervised Model** | |
| CNN | **29.69±2.52** |
| ResNet | 28.55±2.42 |
| LSTM | 26.83±1.32 |
| **Pretrained RNA Language Model** | |
| RNA-FM | 31.62±1.16 |
| RNABERT | 29.77±3.98 |
| RNA-MSM | **34.92±1.99** |
| SpliceBERT-H510 | 26.61±1.30 |
| SpliceBERT-MS510 | 27.13±0.27 |
| SpliceBERT-MS1024 | 27.59±4.61 |
| UTR-LM-MRL | 28.49±1.37 |
| UTR-LM-TE&EL | 32.49±4.14 |
| UTRBERT-3mer | 29.92±1.95 |
| UTRBERT-4mer | 23.20±1.10 |
| UTRBERT-5mer | 25.74±0.00 |
| UTRBERT-6mer | 28.60±1.55 |
| **Our BEACON-B** | |
| BEACON-B | 26.01±1.81 |
| BEACON-B512 | **28.17±1.81** |

### A.2.12 CRISPR On-Target Prediction

For the processing of the on-target sgRNA dataset, we sourced experimentally validated sgRNAs targeting approximately 1,071 genes across four distinct cell lines [19], namely hct116, hek293t, hela, and hl60. To ensure the integrity of our dataset, we selected hl60 cell line data and removed redundant entries and restricted our dataset to sgRNAs with direct experimental validation of knockout efficacy, quantified as the log-fold change.

Furthermore, for a balanced and normalized dataset conducive to regression analysis, we employed a collaborative filtering-based normalization approach [19], akin to methodologies used in user-item recommendation systems. We constructed an efficacy matrix where rows represented experiments and columns represented sgRNAs. The knockout efficacy was normalized by calculating the mean values across the rows, columns, and the entire matrix, and then adjusting the sgRNA efficacy scores by subtracting these mean values and dividing by the number of mean types considered. This normalization process ensures that our dataset remains unbiased and reflective of true knockout efficacies across various experimental setups, thereby enhancing the generalizability and accuracy of our predictive models.

More baselines on CRI-On are shown in Table A.2.11.

Table 13: Comparison on CRI-Off.

| Method | Metric (SC %) |
|---|---|
| **Literature SOTA** | |
| CFD [30] | 10.3 |
| MIT [47] | 8.0 |
| CCTop [98] | 5.2 |
| CROP-IT [96] | 4.2 |
| DeepCRISPR | **12.6** |
| **Naive Supervised Model** | |
| CNN | 11.40±0.10 |
| ResNet | **11.50±0.22** |
| LSTM | 8.60±0.13 |
| **Pretrained RNA Language Model** | |
| RNA-FM | 2.49±1.56 |
| RNABERT | 4.27±1.05 |
| RNA-MSM | 3.85±0.99 |
| SpliceBERT-H510 | 4.00±1.13 |
| SpliceBERT-MS510 | 3.49±2.12 |
| SpliceBERT-MS1024 | **5.00±0.71** |
| UTR-LM-MRL | 4.28±0.15 |
| UTR-LM-TE&EL | 2.91±1.18 |
| UTRBERT-3mer | 4.48±1.12 |
| UTRBERT-4mer | 3.11±1.10 |
| UTRBERT-5mer | 3.93±0.24 |
| UTRBERT-6mer | 4.90±0.57 |
| **Our BEACON-B** | |
| BEACON-B | **4.42±0.33** |
| BEACON-B512 | 3.82±1.04 |

### A.2.13 CRISPR Off-Target Prediction

We followed the dataset from DeepCRISPR [19] including off-target profiles from two distinct cell types: 293-related cell lines and K562 cells, encompassing 30 sgRNAs in total. Using the bowtie2 tool [19], we used K562 cell data and identified approximately 160,000 potential off-target loci across the genome, allowing for up to six nucleotide mismatches per sgRNA. This resulted in a highly unbalanced dataset, with varying numbers of loci associated with each level of mismatch, from one to six.

To address the imbalance and refine our dataset for the regression models, we labeled the off-target sites and normalized them according to the indel frequency detected by various genome-wide off-target detection techniques. This normalization process helped to mitigate the skewness introduced by the uneven distribution of loci, ensuring that our model could generalize effectively across different genomic backgrounds and detection assays.

More baselines on CRI-Off are shown in Table A.2.12.

### A.3 Methods in Benchmark

All pre-trained benchmarked methods use a Masked Language Modeling (MLM) objective.

In the MLM task, a sequence is provided as input, with 15% of its tokens randomly masked. The entire masked sequence is then processed by the model, which is tasked with predicting the original tokens. This approach is analogous to the Cloze task in traditional language modeling.

- 15% of the tokens in the sequence are masked.

- In 80% of the cases, the masked tokens are replaced by a special <mask> token.

- In 10% of the cases, the masked tokens are substituted with a random token different from the original.

- In the remaining 10% of cases, the masked tokens remain unchanged.

### A.3.1 RNABERT

**Training Objectives**   RNABERT was pre-trained with two objectives: masked language modeling (MLM) and structural alignment learning (SAL).

For SAL, the model learns to predict the structural alignment between two RNA sequences. It achieves this by being trained to predict the alignment score of RNA sequence pairs using the Needleman-Wunsch algorithm.

**Training Data**   The RNABERT model was pre-trained using a subset of 76,237 human ncRNA sequences from RNAcentral. The dataset was preprocessed by applying 10 different masking patterns to the 76,237 sequences, resulting in a final dataset comprising 762,370 sequences.

### A.3.2 RNA-FM

**Training Data**   The RNA-FM model was pre-trained using data from RNAcentral. To ensure the dataset was non-redundant, RNA-FM applied CD-HIT (CD-HIT-EST) with a cut-off at 100% sequence identity, resulting in a final dataset containing 23.7 million unique RNA sequences.

### A.3.3 RNA-MSM

Unlike other methods, RNA-MSM utilizes homologous sequences as input to provide additional evolutionary information, similar to MSATransformer [79].

To ensure fairness, homologous sequences were not included in the input during evaluation.

**Training Data**   RNA-MSM was pre-trained using data from Rfam, which includes homologous sequences. To prevent potential overfitting in structural inference, RNA-MSM excluded families with experimentally determined structures, such as ribosomal RNAs, transfer RNAs, and small nuclear RNAs. The final dataset comprises 3,932 RNA families, with a median of 2,184 MSA sequences per family. To augment the number of homologous sequences, RNA-MSM employed an automated pipeline, RNAcmap3 [16], for homolog search and sequence alignment.

### A.3.4 SpliceBERT

**Training Data**   The SpliceBERT model was pre-trained using messenger RNA precursor sequences obtained from the UCSC Genome Browser.

SpliceBERT gathered reference genomes and gene annotations from the UCSC Genome Browser for 72 vertebrate species. Bedtools getfasta was used to extract pre-mRNA sequences from the reference genomes based on these gene annotations. The resulting pre-mRNA sequences were then utilized for pre-training SpliceBERT. The pre-training dataset comprises 2 million pre-mRNA sequences, with a total length of 65 billion nucleotides.

### A.3.5 3UTRBERT

**Training Data**   The 3UTRBERT model was pre-trained using human mRNA transcript sequences obtained from GENCODE.

3UTRBERT collected 108,573 unique human mRNA transcripts from GENCODE, utilizing only the longest transcript for each gene in the pre-training process. To avoid codon constraints in the CDS region and to reduce the complexity of the full mRNA transcripts, only the 3' untranslated regions (3'UTRs) of the mRNA transcripts were used. The average length of the 3'UTRs was 1,227 nucleotides, with a median length of 631 nucleotides. Each 3'UTR sequence was divided into non-overlapping patches of 510 nucleotides, with the remaining sequences padded to the same length.

### A.3.6 UTR-LM

**Training Objectives**   In addition to MLM pre-training, UTR-LM employs two additional supervised objectives: Secondary Structure (SS) and Minimum Free Energy (MFE).

Both secondary structure and the MFE value are calculated using ViennaRNA [67]. To prevent information leakage, UTR-LM calculates the secondary structure loss only on the masked positions. The output embedding of the cls token is used by UTR-LM to regress the MFE value.

**Training Data**   The UTR-LM model was pre-trained using 5' UTR sequences sourced from three origins: the Ensembl database, synthetic libraries from Sample et al. [90], and endogenous human 5' UTR data analyzed by Cao et al. [10].

The preprocessing of 5' UTR sequences for UTR-LM involved a 4-step pipeline: First, all coding sequences (CDS) and non-5' UTR fragments were removed from the raw sequences. Second, duplicate sequences were identified and removed. Third, the sequences were truncated to fit within a range of 30 to 1022 base pairs. Finally, incorrect and low-quality sequences were filtered out.

### A.3.7 BEACON-B

**Training Data**   We filter 523,934 human ncRNA sequences from the total ncRNA in the RNACentral database [104] as pre-training data. BEACON-B and BEACON-B512 use normal BERT-base [28] architecture with 12 layers.

The pre-training configs of BEACON-B and BEACON-B512 are shown as Table 14 and Table 15.

Table 14: Configuration settings for the BEACON-B pre-training.

| Config | Value |
|---|---|
| optimizer | AdamW |
| optimizer epsilon | 1e-6 |
| optimizer momentum | $\beta_1, \beta_2 = 0.9, 0.98$ |
| weight decay | 0.01 |
| learning rate sch. | linear decay |
| learning rate | 2e-4 |
| warmup steps | 10000 |
| steps | 80000 |
| batch size | 256 |
| gradient accumulation | 2 |
| dtype | float16 |
| length | 1026 |
| pertaining data | RNACentral Human ncRNA |

### A.4 Computational Resources

We fine-tune or train each model from scratch on one task using one NVIDIA A100 40g GPU. We pre-train the simple BEACON-B on 8 A100 GPUs of 80GB for 1.3 days and BEACON-B512 on 4 A100 GPUs of 80GB for 0.895 days.

### A.5 Computational Complexity Comparisons

We have compared the currently available pre-training resource consumption of RNA language models in Table 5, shown via GPU Days (Days * GPUs). We also compared the FLOPs and MACs of the different models in Table 16. The detailed computational costs of the models are outlined as follows (following the sequence lengths at which the models were pretrained). Despite our model employing a vanilla BERT-base model with 12 layers, which doesn't minimize FLOPs, it still manages to achieve the lowest resource usage, requiring only 3.58 GPU Days. This efficiency is primarily due to our use of a modestly sized, filtered ncRNA dataset combined with a lightweight pre-training schedule (80K steps and a batch size of 512).

Table 15: Configuration settings for the BEACON-B512 pre-training.

| Config | Value |
|---|---|
| optimizer | AdamW |
| optimizer epsilon | 1e-6 |
| optimizer momentum | $\beta_1, \beta_2 = 0.9, 0.98$ |
| weight decay | 0.01 |
| learning rate sch. | linear decay |
| learning rate | 2e-4 |
| warmup steps | 10000 |
| steps | 80000 |
| batch size | 512 |
| gradient accumulation | 1 |
| dtype | float16 |
| length | 512 |
| pertaining data | RNACentral Human ncRNA |
| attention | FlashAttention |

Table 16: FLOPs, MACs, and GPU days for various models.

| Method | FLOPs (G) | MACs (G) | GPU Days (Days * GPUs) |
|---|---|---|---|
| RNA-FM | 233.88 | 116.77 | 30 * 8 = 240 |
| RNABERT | 0.91 | 0.46 | - |
| RNA-MSM | 201.46 | 84.56 | - |
| SpliceBERT-H510 | 19.27 | 9.63 | 7 * 8 = 56 |
| SpliceBERT-MS510 | 19.27 | 9.63 | 7 * 8 = 56 |
| SpliceBERT-MS1024 | 38.69 | 19.33 | 7 * 8 + 3 * 4 = 68 |
| UTR-LM-MRL | 5.76 | 2.83 | - |
| UTR-LM-TE&EL | 5.76 | 2.83 | - |
| UTRBERT-3mer | 511.41 | 255.56 | 38 * 4 = 152 |
| UTRBERT-4mer | 511.41 | 255.56 | 38 * 4 = 152 |
| UTRBERT-5mer | 511.41 | 255.56 | 38 * 4 = 152 |
| UTRBERT-6mer | 511.41 | 255.56 | 38 * 4 = 152 |
| BEACON-B | 198.48 | 99.13 | 1.3 * 8 = **10.4** |
| BEACON-B512 | 87.02 | 43.49 | 0.895 * 4 = **3.58** |

## A.6 Additional Results

For the experiments 4 on component analysis of the baseline RNA language model, we further counted the number of top performances for different tokenizers and positional encoding as shown in Table 17. The effectiveness of Single nucleotide tokenizer and ALiBi can be demonstrated directly.

In addition, we further collected additional testing datasets to evaluate BEACON-B and other models for their generalization. On the one hand, we collected the contact map data W19 from Rfam [112], results with direct generalization are outlined in Table A.6. It shows our BEACON-B models can achieve significant performance.

On the other hand, we collected two unseen APA isoform libraries HSPE1 and WHAMMP2 from MPRAs [94, 9], results with direct generalization are outlined in Table 19. On the task of predicting 3'UTR functions, BEACON-B's generalization capabilities show potential to rival those of UTRBERT-3mer, which was specifically pre-trained on 3'UTR sequences.

## A.7 Ablation Study

We carried out detailed ablation experiments in the pre-training setting in Table 20. We used the same pre-training setup of BEACON-B512 (pre-training on 512-length sequences of human-filtered ncRNA and using FlashAttn). This setup allowed us to scrutinize the impact of various tokenizers and positional encodings. For downstream evaluation, in order to minimize computational resources and accelerate experimental procedures, we evaluated three major RNA task categories: Distance

Table 17: The number of Top2 performance among different tokenizers and positional encodings.

| Rank | Tokenizer | | | | Positional encoding | | |
|------|-----------|------|-------------|-----|-----|-------|------|
|      | Single | 6mer | Non-overlap | BPE | APE | ALiBi | RoPE |
| 1st  | 11 | 0 | 1 | 1 | 5 | 7 | 1 |
| 2nd  | 1  | 6 | 3 | 3 | 8 | 5 | 0 |

Table 18: Additional Results on W19 Dataset of CMP.

| Method | Metric (SC %) |
|--------|---------------|
| PLMC | 48.00 |
| RNAcontact | 72.00 |
| RNA-FM | 55.98 |
| RNABERT | 26.19 |
| SpliceBERT-H510 | 66.00 |
| UTR-LM-MRL | 60.00 |
| UTRBERT-3mer | 22.27 |
| UTRBERT-6mer | 53.41 |
| BEACON-B | **76.01** |
| BEACON-B512 | 73.56 |

Map Prediction (DMP) from structural tasks, APA Isoform Prediction (APA) from functional tasks, and Vaccine Degradation Prediction (VDP) from engineering tasks.

## A.8 More Discussion of Future Work and Limitations

We provide additional discussions with the following two aspects:

**RNA tasks with other inputs:** To ensure fairness in comparison, we standardize input formats by using RNA sequences across all tasks and all models. This approach was necessary to maintain consistency in evaluations. However, it may not fully capture the potential of models that are designed to leverage richer input data, such as the scRNA-seq data for cell annotation and gene regulation network which uses genes with expression level [105] and inverse RNA folding using structure as input [46]. Future work could explore alternative evaluation settings that allow for the use of more complex inputs, thereby providing a more nuanced assessment of model performance.

In particular, although the task of inverse RNA folding extends beyond the current scope of this benchmark, we plan to extend our benchmark to support more complex tasks involving diverse types of input, broadening our coverage in RNA research significantly in the future. Specifically, for inverse RNA folding, we will consider datasets and tasks such as Eterna100v1 [4] and Eterna100v2 [59] for standard inverse RNA folding evaluations and PseudoBase++ [106] to evaluate the capability of inverse folding extending to pseudoknots. Regarding models, we will evaluate significant inverse RNA folding methods such as RNAinverse [46], MCTS-RNA [118], LEARNA [86], MetaLEARNA [86], gRNAde [56], and RiboDiffusion [49]. We are also considering influential inverse protein folding models like StructGNN [52], GVP-GNN [55], and ProteinMPNN [27], PiFold [37], given their potential to offer new insights into RNA structure modeling, owing to the inverse task between

Table 19: Additional Results on HSPE1 and WHAMMP2 Dataset of APA.

| Method | HSPE1 Metric ($R^2$ %) | WHAMMP2 Metric ($R^2$ %) |
|--------|------------------------|--------------------------|
| APARENT | 32.95 | 18.56 |
| RNA-FM | 35.35 | 16.43 |
| RNABERT | 19.04 | 3.43 |
| SpliceBERT-H510 | 17.58 | 4.81 |
| UTR-LM-MRL | 28.62 | 8.73 |
| UTRBERT-3mer | 33.73 | 17.56 |
| UTRBERT-6mer | 42.71 | 20.17 |
| BEACON-B | 29.98 | 15.35 |
| BEACON-B512 | 33.22 | 17.83 |

Table 20: Performance comparison of different methods on CMP, APA, and VDP tasks.

| Method | Tokenizer | Positional Encoding | Metric (P@L %) | Metric ($R^2$ %) | Metric (MCRMSE↓) |
|---|---|---|---|---|---|
| BPE-ALiBi | BPE | ALiBi | 49.08 | 64.98 | 0.640 |
| Non-overlap-ALiBi | Non-overlap | ALiBi | 47.68 | 66.61 | 0.639 |
| 6mer-ALiBi | 6mer | ALiBi | 48.79 | 70.77 | 0.372 |
| Single-RoPE | Single | RoPE | 12.84 | 33.86 | 0.462 |
| Single-APE | Single | APE | 50.76 | 66.20 | 0.327 |
| BEACON-B512 | Single | ALiBi | **56.82** | **72.00** | **0.320** |

modeling protein and RNA structures. We could also explore using cross-modal generation methods [2, 64, 81, 80] to bridge the structure and sequence for the inverse task. For evaluation metrics, we could consider assessing sequence similarity, sequence diversity, and structural similarity, using indicators such as recovery rate, diversity, and structural similarity.

**Other important RNA sequence tasks:** While our current benchmark effectively evaluates RNA language models across a spectrum of tasks, we acknowledge the out-of-selection of certain complex tasks such as 3D RNA structure prediction. This is primarily due to several challenges: (1) The majority of open-source RNA language models have not been developed to perform 3D structure prediction tasks. (2) The acquisition of high-quality structural data, often requiring Multiple Sequence Alignment (MSA) data, is prohibitively expensive. This, coupled with the limited availability of such data, poses a significant barrier [108]. (3) The computational pipeline for predicting 3D structures (e.g., structural modules) is exceedingly complex and resource-intensive, often necessitating days of computation on high-performance GPUs [65]. In future extensions of our benchmark, we aim to incorporate more diverse tasks, including those requiring complex data inputs and significant computational resources, to better encompass the breadth of challenges in RNA research.

## A.9 Broader Societary Impacts

This work is dedicated to establishing a robust and versatile benchmark for RNA-related tasks, enhancing the understanding of RNA sequences across diverse applications. Our benchmark, encompassing a variety of RNA tasks, aims to rigorously evaluate the efficacy of different RNA representations covering structural analysis, functional studies, and engineering applications. By doing so, it provides a critical assessment of their potential utility in real-world scenarios, thereby laying a foundational framework for applying deep learning in fields such as medical research and genetics.

However, it is also important to acknowledge the dual-use nature of any powerful technology [48, 122, 31], including those developed from our benchmark. For instance, the enhanced ability to manipulate RNA sequences might be misused, such as in the creation of adverse viral agents. Moving forward, it is crucial to address these risks. We will develop and implement guidelines for the ethical and safe use of our benchmark in the future, ensuring that it contributes positively to society and does not enable harmful applications.

## A.10 Assets

Table 21: Software used in this work

| Asset | License |
|---|---|
| FlashAttention [26, 25] | BSD-3-Clause |
| Pytorch [8] | BSD-3-Clause |
| Pytorch Lightning [33] | Apache-2.0 |
| Huggingface [113] | Apache-2.0 |
| Scikit-Learn [75] | BSD-3-Clause |
| Numpy [43] | BSD-3-Clause |
| Matplotlib [51] | Matplotlib License |
| Seaborn [109] | Apache-2.0 |

### A.10.1 Software and Libraries

The open-source software, and corresponding licenses are presented in Table 21. The data, licenses and corresponding URL are presented in Table 22.

Table 22: Dataset used in this work

| Dataset | Sub-dataset | License | URL |
|---|---|---|---|
| bpRNA-1M | CRW | - | `https://bprna.cgrb.oregonstate.edu/about.php` `https://crw-site.chemistry.gatech.edu` |
| | tmRDB | Research Purpose Only | `https://rth.dk/resources/rnp/tmRDB/` |
| | SRPDB | Research Purpose Only | `https://rth.dk/resources/rnp/SRPDB/` |
| | tRNADB | | |
| | Rnase P | Public Domain | |
| | RFam | CC0 1.0 | `https://rfam.org` |
| | PDB | CC0 1.0 | `https://www.rcsb.org` |
| RNAcontact | | Public Domain | `https://yanglab.qd.sdu.edu.cn/RNAcontact/` |
| StructImpute | | MIT & Non Commerical | `https://figshare.com/articles/dataset/A_deep_learning_method_for_recovering_missing_signals_in_transcriptome-wide_RNA_structure_profiles_from_probing_experiments/16606850` |
| SpliceAI | | GPLv3 | `https://github.com/illumina/SpliceAI` |
| APARNET | | MIT | `https://github.com/johli/aparent` |
| ncRNA | | Apache 2.0 | `https://github.com/bioinformatics-sannio/ncrna-deep` |
| MultiRM | | MIT | |
| Optimus | | GEO | `https://www.ncbi.nlm.nih.gov/geo/query/acc.cgi?acc=GSE114002` |
| OpenVaccine | | Non Commerical | `https://www.kaggle.com/competitions/stanford-covid-vaccine/data` |
| ProgrammableRNAswitches | | GEO | `https://www.ncbi.nlm.nih.gov/geo/query/acc.cgi?acc=GSE149225` |
| DeepCRISPR | | Apache 2.0 | |

