# OpenReview forum: "BEACON: Benchmark for Comprehensive RNA Tasks and Language Models"
_NeurIPS.cc/2024/Datasets_and_Benchmarks_Track — NeurIPS 2024 Track Datasets and Benchmarks Poster_

### Official Review · Reviewer_xw6V · 2024-06-30

**Rating:** 8
**Confidence:** 2
**Clarity:** Yes, the paper is well-written and cl…

**Review:**

Pros:

- See strengths below.

Cons:

- This work lacks an explanation for why these particular language models were chosen. Is it solely because they are "capable of tackling multiple RNA-related tasks by leveraging advanced language modeling techniques"? Are there other applicable language models that were considered?
- In the component analysis, tokenization and positional encoding techniques are combined with the baseline RNA LM. The introduction to the baseline RNA LM is missing, which could be problematic for those unfamiliar with the field.

**Strengths:**

- This work proposes the first comprehensive RNA benchmark, including 13 tasks (with detailed explanations and evaluation metrics) across different fields. This benchmark will be beneficial for evaluating RNA language models.
- This work presents the evaluation of existing models and identifies two key components of RNA language models, which will be advantageous for future RNA model development.

**Additional Feedback:**

NA

**Correctness:**

The dataset is constructed soundly, and the evaluation methods are performed correctly.

**Documentation:**

The documentation provided is sufficient.

**Ethics:**

No ethical concerns

**Limitations:**

Yes

**Opportunities For Improvement:**

- Provide an introduction to the baseline RNA LM.
- Justify the selection of these language models.
- While BEACON-B is tested on the proposed comprehensive BEACON benchmark, additional testing on other datasets would enhance its credibility.
- Explain "P@L" and "ACC@K" in Table 3, despite being found in Section 3.
- The text color for "first", "second", "third", and "fourth" is too light. Use black text with different colored backgrounds, as in the table.
- In academic writing, it's uncommon to use abbreviations for "Table". Ensure consistency: some are abbreviated as "Tab" and some as "Tab. ".

**Relation To Prior Work:**

Yes, the authors adequately discussed the relation to prior work. See Section 2.

**Summary And Contributions:**

Summary:

This paper introduces BEACON, a comprehensive RNA benchmark designed to assess the effectiveness of RNA models. Various RNA models were evaluated, and a baseline method, BEACON-B, was proposed.

Contribution:

- The benchmark integrates datasets from 13 RNA tasks across three fields: structural analysis, functional studies, and engineering applications.
- It evaluates models, including naive supervised models and pre-trained language models, and identifies key components of these language models.
- It proposes BEACON-B, a robust yet cost-effective baseline method for RNA analysis.

---

> ### Author Rebuttal · Authors · 2024-08-17
>
> We sincerely thank the reviewer for the insightful feedback. Below are our responses to the points raised:
>
> ##  O1: Introduction to Baseline RNA LM
>
> Baseline RNA LMs include 12 baseline models without pre-training for component analysis and 2 baseline pre-trained models (BEACON-B and BEACON-B512).
>
> We will highlight this section in the main text to ensure that readers are aware of the baseline RNA LM and BEACON-B.
>
> The 12 baseline models are different combinations of 4 tokenizers: non-overlapping 6mer (Non-overlap), byte-pair encoding (BPE), single nucleotide (Single), overlapping 6mer (6mer), and 3 positional encodings: learned absolute position embedding (APE), rotary position embedding (RoPE), attention with linear bias (ALiBi). Each model out of the 12 is the combination of one tokenizer and one positional encoding. As for other components, it largely follows a vanilla BERT-base model with 12 layers, 768 hidden units, and 12 attention heads. It is trained directly on downstream tasks without any pre-training on RNA sequences.
>
> The BEACON-B and BEACON-B512 models use Single tokenization and ALiBi positional encoding, as suggested by the ablation studies of the 12 baseline models. They use the same model architecture as the baseline RNA LM, but are pre-trained on 523,934 human ncRNA sequences filtered from the RNACentral701 database, with a detailed pre-training setting provided in Appendix A.2.7. They only differ in the sequence length used for pre-training: BEACON-B is pre-trained on 1026-length sequences, while BEACON-B512 is pre-trained on 512-length sequences.
>
> We will highlight this section in the main text to ensure that readers are aware of the baseline RNA LM and BEACON-B.
>
> ## O2: Justification for the Selection of Language Models
>
> We have included all RNA language models that had open-source weights available at the time of our work and that were pre-trained on RNA sequences using Masked Language Modeling (MLM). Since we did not perform any specific model selection, we refer to these models as "capable of tackling multiple RNA-related tasks by leveraging advanced language modeling techniques," which is a common characteristic of pre-trained language models. We will clarify this in the main text to avoid any confusion.
>
> ## O3: Additional Testing on Other Datasets
>
> Thanks for your suggestion. Our primary aim is to introduce a comprehensive benchmark for evaluating RNA language models. The BEACON benchmark, which we have developed, incorporates a diverse range of tasks and datasets, offering a robust framework for evaluation. Given the breadth and thoroughness of this benchmark, we believe that the current benchmark already provides a solid and broad evaluation of RNA language models.
>
> In addition, we further collected additional testing datasets to evaluate BEACON-B and other models for their generalization. On the one hand, we collected the contact map data from Rfam [r1], results with direct generalization are outlined as follows. It shows our BEACON-B models can achieve significant performance.
>
> |        Method       | Metric (P@L %) |
> |:---------------|:--------------:|
> |         PLMC        |       48       |
> |      RNAcontact     |       72       |
> |        RNA-FM       |      55.98     |
> |       RNABERT       |      26.19     |
> | SpliceBERT-H510     |      66.00     |
> |      UTR-LM-MRL     |      60.00     |
> |     UTRBERT-3mer    |      22.27     |
> |     UTRBERT-6mer    |      53.41     |
> |       BEACON-B      |      76.01     |
> |     BEACON-B512     |      73.56     |
>
>
> On the other hand, we collected two unseen APA isoform libraries HSPE1 and WHAMMP2 from MPRAs [r2, r3], results with direct generalization are outlined as follows. On the task of predicting 3'UTR functions, BEACON-B's generalization capabilities show potential to rival those of UTRBERT-3mer, which was specifically pre-trained on 3’UTR sequences.
>
> |      Method     |      HSPE1     |     WHAMMP2    |
> |:---------------|:--------------:|:--------------:|
> |                 | Metric (R^2 %) | Metric (R^2 %) |
> |     APARENT     |      32.95     |      18.56     |
> |      RNA-FM     |      35.35     |      16.43     |
> |     RNABERT     |      19.04     |      3.43      |
> | SpliceBERT-H510 |      17.58     |      4.81      |
> |    UTR-LM-MRL   |      28.62     |      8.73      |
> |   UTRBERT-3mer  |      33.73     |      17.56     |
> |   UTRBERT-6mer  |      42.71     |      20.17     |
> |     BEACON-B    |      29.98     |      15.35     |
> |   BEACON-B512   |      33.22     |      17.83     |
>
> [r1] Weinreb, C., Riesselman, A. J., Ingraham, J. B., Gross, T., Sander, C., & Marks, D. S. (2016). 3D RNA and functional interactions from evolutionary couplings. Cell, 165(4), 963-975.
>
> [r2] Shigaki, D., Adato, O., Adhikari, A. N., Dong, S., Hawkins‐Hooker, A., Inoue, F., ... & Beer, M. A. (2019). Integration of multiple epigenomic marks improves prediction of variant impact in saturation mutagenesis reporter assay. Human mutation, 40(9), 1280-1291.
>
> [r3] Bogard, N., Linder, J., Rosenberg, A. B., & Seelig, G. (2019). A deep neural network for predicting and engineering alternative polyadenylation. Cell, 178(1), 91-106.

---

> > ### Author Rebuttal · Authors · 2024-08-17
> >
> > ## O4: Explanation of "P@L" and "ACC@K"
> >
> > We have revised the caption of Table 3 for clarity:
> >
> > Table 3: Benchmark results across 13 RNA-related tasks. We use four shades of blue to denote the first, second, third, and fourth best performances among naive supervised models and pre-trained RNA LMs. Mean (std) is reported for each experiment. P@L represents Top-L precision, and ACC@K denotes Top-K accuracy.
> >
> > Additionally, we will include the precise definition of Top-L precision and Top-K accuracy in section 3 as well. The definitions of them are as follows:
> >
> > The top-k accuracy for a particular class is defined as follows: Suppose the test set has k positions that belong to the class. We choose the threshold so that exactly k test set positions are predicted as belonging to the class. The fraction of these k predicted positions that truly belong to the class is reported as the top-k accuracy.
> >
> > The top-L precision for a particular class is defined as follows: Suppose the length of a sample sequence is L. We choose the threshold so that exactly L positions are predicted as belonging to the class. The fraction of these L-predicted positions that truly belong to the class is reported as the top-L precision.
> >
> > ## O5: Text Color for "First," "Second," "Third," and "Fourth"
> >
> > Thank you for this suggestion to improve table readability. We will change the text color for "first," "second," "third," and "fourth" to black with different colored backgrounds in the final version.
> >
> > ## O6: Abbreviations for "Table"
> >
> > We appreciate you pointing out this issue. We will use "Table" instead of abbreviations in the final version for consistency.

---

> > > ### Comment · Reviewer_xw6V · 2024-08-30
> > >
> > > Thanks to the authors for addressing my concerns. I've raised the score accordingly.

---

> > > > ### Author Response · Authors · 2024-08-30
> > > > **Thanks for the response**
> > > >
> > > > Thank you very much for your support and for increasing your score to 8. We’re glad that the explanations and additional testing experiments have resolved your concerns. We will incorporate these detailed explanations and experiments into the final version of the paper. If you have any further suggestions or concerns, please feel free to reach out. We are always receptive to feedback and dedicated to enhancing the quality of our work.

---

### Official Review · Reviewer_Wr93 · 2024-07-25

**Rating:** 6
**Confidence:** 3
**Correctness:** Seems correct

**Review:**

Please see Strengths and Opportunities For Improvement.

**Strengths:**

1. BEACON covers a wide range of RNA-related tasks, providing a holistic evaluation framework.

2. The authors assess various models, including traditional approaches and RNA language models. The paper provides valuable insights into the components of RNA language models.

3. The proposed BEACON-B model offers an efficient baseline for future research in RNA modeling.

**Additional Feedback:**

Typo in line 272: "cotains" should be "contains"

**Clarity:**

Overall well written. The current experiment tables are hard to read and can be improved with performance rank ordering.

**Documentation:**

Documentation provided for most of the code. Can be improved with more usage guide.

**Limitations:**

The paper includes discussions on limitations.

**Opportunities For Improvement:**

1. Limited baselines. While the paper evaluates several models, it could benefit from including more recent baselines in specific tasks. For example, in the RNA secondary structure prediction task, including methods like E2Efold or SPOT-RNA2 would strengthen the comparison.

2. Can authors justify how to ensure testing data are not present among training data?

3. The paper doesn't provide detailed ablation studies for the proposed BEACON-B model, which would help understand the actual contribution of each component.

4. The paper can benefit from a detailed discussion on how the datasets for each task were preprocessed or balanced, which could affect model performance.

5. While the paper mentions the GPU used for training and fine-tuning, it can benefit from computational complexity comparisons across models/methods.

**Relation To Prior Work:**

The paper includes comparisons with prior works

**Summary And Contributions:**

This paper introduces BEACON, a benchmark for RNA-related tasks. It encompasses 13 tasks covering structural analysis, functional studies, and engineering applications of RNA. The authors evaluate various models, including traditional approaches like CNNs and advanced RNA foundation models, on these tasks. They also analyze key components of RNA language models, focusing on tokenization methods and positional encodings. Based on their findings, they propose BEACON-B, an efficient baseline model incorporating single nucleotide tokenization and Attention with Linear Biases.

---

> ### Author Rebuttal · Authors · 2024-08-17
>
> We thank the reviewer for the valuable and insightful feedback, and here we provide corresponding responses to address these concerns.
>
> ## O1: More Baselines in Specific Tasks
> Our benchmark focuses primarily on evaluating RNA language models, also comparing various baseline methods. We have compared the performance with one best-performing state-of-the-art (SOTA) method for the specific task on each dataset in the paper. In addition, as per your nice suggestion, for tasks that feature more significant or recently established specific SOTA baseline models, we have included these in "Literature SOTA" for comparison as well.
>
> - **Secondary Structure Prediction**
>
>     |             Method            | Metric (F1 %) |
>     |:-----------------------------:|:-------------:|
>     |         **Literature SOTA**                        |
>     |          Contextfold [r1]          |      54.6     |
>     |           CONTRAfold [r2]          |      56.7     |
>     |            E2Efold [r3]           |       13      |
>     |          RNAstructure [r4]        |      53.3     |
>     |            RNAsoft [r5]           |      53.5     |
>     |            RNAfold [r6]           |      53.6     |
>     |             Mfold [r7]            |      53.8     |
>     |           LinearFold [r8]         |       55      |
>     |            MXfold2 [r9]           |      55.8     |
>     |          Externafold [r10]         |      56.3     |
>     |            SPOT-RNA [r11]          |      61.9     |
>     |             UFold             |      **65.4**     |
>     |       **Naive Supervised Model**      |                |
>     |                CNN                |   49.95±0.82   |
>     |               ResNet              | 57.26±3.14 |
>     |                LSTM               |   **58.61±0.21**   |
>     | **Pretrained RNA Language Model**                |
>     |             RNA-FM            |   **68.50±0.54**  |
>     |            RNABERT            |   57.27±0.30  |
>     |            RNA-MSM            |   57.98±0.47  |
>     |      SpliceBERT-H510      |   64.93±0.84  |
>     |        SpliceBERT-MS510       |  43.24±28.64  |
>     |       SpliceBERT-MS1024       |   68.26±0.20  |
>     |           UTR-LM-MRL          |   59.71±0.30  |
>     |          UTR-LM-TE&EL         |   59.57±0.20  |
>     |          UTRBERT-3mer         |   60.37±0.47  |
>     |          UTRBERT-4mer         |   59.41±0.45  |
>     |          UTRBERT-5mer         |   47.92±8.75  |
>     |          UTRBERT-6mer         |  38.56±28.76  |
>     |            **Our BEACON-B**           |                |
>     |            BEACON-B           |   **64.18±0.44**  |
>     |          BEACON-B512          |   58.75±3.72  |
>
> [r1] Zakov, S., Goldberg, Y., Elhadad, M., & Ziv-Ukelson, M. (2011). Rich parameterization improves RNA structure prediction. Journal of Computational Biology, 18(11), 1525-1542.
>
> [r2] Do, C. B., Woods, D. A., & Batzoglou, S. (2006). CONTRAfold: RNA secondary structure prediction without physics-based models. Bioinformatics, 22(14), e90-e98.
>
> [r3] Chen, X., Li, Y., Umarov, R., Gao, X., & Song, L. (2020). RNA secondary structure prediction by learning unrolled algorithms. arXiv preprint arXiv:2002.05810.
>
> [r4] Mathews, D. H., & Turner, D. H. (2006). Prediction of RNA secondary structure by free energy minimization. Current opinion in structural biology, 16(3), 270-278.
>
> [r5] Andronescu, M., Aguirre-Hernandez, R., Condon, A., & Hoos, H. H. (2003). RNAsoft: a suite of RNA secondary structure prediction and design software tools. Nucleic acids research, 31(13), 3416-3422.
>
> [r6] Lorenz, R., Bernhart, S. H., Höner zu Siederdissen, C., Tafer, H., Flamm, C., Stadler, P. F., & Hofacker, I. L. (2011). ViennaRNA Package 2.0. Algorithms for molecular biology, 6, 1-14.
>
> [r7] Zuker, M. (2003). Mfold web server for nucleic acid folding and hybridization prediction. Nucleic acids research, 31(13), 3406-3415.
>
> [r8] Huang, L., Zhang, H., Deng, D., Zhao, K., Liu, K., Hendrix, D. A., & Mathews, D. H. (2019). LinearFold: linear-time approximate RNA folding by 5'-to-3'dynamic programming and beam search. Bioinformatics, 35(14), i295-i304.
>
> [r9] Sato, K., Akiyama, M., & Sakakibara, Y. (2021). RNA secondary structure prediction using deep learning with thermodynamic integration. Nature communications, 12(1), 941.
>
> [r10] Wayment-Steele, H. K., Kladwang, W., Strom, A. I., Lee, J., Treuille, A., Becka, A., ... & Das, R. (2022). RNA secondary structure packages evaluated and improved by high-throughput experiments. Nature methods, 19(10), 1234-1242.
>
> [r11] Singh, J., Hanson, J., Paliwal, K., & Zhou, Y. (2019). RNA secondary structure prediction using an ensemble of two-dimensional deep neural networks and transfer learning. Nature communications, 10(1), 5407.

---

> > ### Author Rebuttal · Authors · 2024-08-17
> >
> > - **Contact Map Prediction**
> >
> >     |               Method              |  Metric (P@L %) |
> >     |:---------------------------------:|:--------------:|
> >     |         **Literature SOTA**        |                |
> >     |                PLMC [r12]               |       28       |
> >     |          RNAcontact (Seq) [r13]         |       33       |
> >     |          RNAcontact (Cov) [r13]        |       59       |
> >     |          RNAcontact (SS)  [r13]        |       58       |
> >     |             RNAcontact            |     **66**     |
> >     |       **Naive Supervised Model**      |                |
> >     |                CNN                |   43.89±5.53   |
> >     |               ResNet              | **59.59±0.68** |
> >     |                LSTM               |   40.41±1.67   |
> >     | **Pretrained RNA Language Model** |                |
> >     |               RNA-FM              |   47.56±6.73   |
> >     |              RNABERT              |   45.21±10.87  |
> >     |              RNA-MSM              |   57.26±15.38  |
> >     |        SpliceBERT-H510        |   45.80±6.03   |
> >     |          SpliceBERT-MS510         |   52.64±7.56   |
> >     |         SpliceBERT-MS1024         |   47.32±3.16   |
> >     |             UTR-LM-MRL            |   45.51±23.51  |
> >     |            UTR-LM-TE&EL           | **60.32±7.27** |
> >     |            UTRBERT-3mer           |   51.03±21.48  |
> >     |            UTRBERT-4mer           |   44.91±27.56  |
> >     |            UTRBERT-5mer           |   44.71±7.64   |
> >     |            UTRBERT-6mer           |   51.56±20.30  |
> >     |            **Our BEACON-B**           |                |
> >     |              BEACON-B             |   60.81±1.70   |
> >     |            BEACON-B512            | **61.20±2.11** |
> >
> >     [r12] Weinreb, C., Riesselman, A. J., Ingraham, J. B., Gross, T., Sander, C., & Marks, D. S. (2016). 3D RNA and functional interactions from evolutionary couplings. Cell, 165(4), 963-975.
> >
> >     [r13] Sun, S., Wang, W., Peng, Z., & Yang, J. (2021). RNA inter-nucleotide 3D closeness prediction by deep residual neural networks. Bioinformatics, 37(8), 1093-1098.
> >
> > - **Non-coding RNA Function Classification**
> >
> >     |               Method              |  Metric (ACC %)  |
> >     |:---------------------------------:|:---------------:|
> >     |         **Literature SOTA**        |                 |
> >     |                EDeN [r14]              |        67       |
> >     |                nRC [r15]               |      81.81      |
> >     |               RNAGCN              |    **85.73**    |
> >     |     **Naive Supervised Model**    |                 |
> >     |                CNN                |    88.62±0.71   |
> >     |               ResNet              |    88.33±1.22   |
> >     |                LSTM               |  **88.78±0.10** |
> >     | **Pretrained RNA Language Model** |                 |
> >     |               RNA-FM              | **96.81±0.061** |
> >     |              RNABERT              |   68.95±7.285   |
> >     |              RNA-MSM              |   84.85±0.266   |
> >     |        SpliceBERT-H510        |   95.92±0.666   |
> >     |          SpliceBERT-MS510         |   95.87±0.364   |
> >     |         SpliceBERT-MS1024         |   96.05±0.777   |
> >     |             UTR-LM-MRL            |   89.97±0.617   |
> >     |            UTR-LM-TE&EL           |   81.33±8.551   |
> >     |            UTRBERT-3mer           |   92.88±0.379   |
> >     |            UTRBERT-4mer           |   94.32±0.946   |
> >     |            UTRBERT-5mer           |   93.04±0.367   |
> >     |            UTRBERT-6mer           |   93.12±0.168   |
> >     |          **Our BEACON-B**         |                 |
> >     |              BEACON-B             |    94.63±0.16   |
> >     |            BEACON-B512            |  **94.99±0.21** |
> >
> >     [r14] Navarin, N., & Costa, F. (2017). An efficient graph kernel method for non-coding RNA functional prediction. Bioinformatics, 33(17), 2642-2650.
> >
> >     [r15] Fiannaca, A., La Rosa, M., La Paglia, L., Rizzo, R., & Urso, A. (2017). nRC: non-coding RNA Classifier based on structural features. BioData mining, 10, 1-18.

---

> > ### Author Rebuttal · Authors · 2024-08-17
> >
> > - **CRISPR On-Target Prediction**
> >
> >     |               Method              |  Metric (SC %) |
> >     |:---------------------------------:|:--------------:|
> >     |         **Literature SOTA**        |                |
> >     |              CHOPCHOP [r16]            |       0.9      |
> >     |              DeepCRISPR [r17]     |      26.2      |
> >     |             WU-CRISPR [r18]            |      26.8      |
> >     |            sgRNA Scorer [r19]          |      30.5      |
> >     |        CRISPR MultiTargeter [r20]      |      32.5      |
> >     |              E-CRISP [r21]             |      33.6      |
> >     |           sgRNA Designer [r22]         |      41.8      |
> >     |                SSC                |    **44.1**    |
> >     |     **Naive Supervised Model**    |                |
> >     |                CNN                | **29.69±2.52** |
> >     |               ResNet              |   28.55±2.42   |
> >     |                LSTM               |   26.83±1.32   |
> >     | **Pretrained RNA Language Model** |                |
> >     |               RNA-FM              |   31.62±1.16   |
> >     |              RNABERT              |   29.77±3.98   |
> >     |              RNA-MSM              | **34.92±1.99** |
> >     |        SpliceBERT-H510        |   26.61±1.30   |
> >     |          SpliceBERT-MS510         |   27.13±0.27   |
> >     |         SpliceBERT-MS1024         |   27.59±4.61   |
> >     |             UTR-LM-MRL            |   28.49±1.37   |
> >     |            UTR-LM-TE&EL           |   32.49±4.14   |
> >     |            UTRBERT-3mer           |   29.92±1.95   |
> >     |            UTRBERT-4mer           |   23.20±1.10   |
> >     |            UTRBERT-5mer           |   25.74±0.00   |
> >     |            UTRBERT-6mer           |   28.60±1.55   |
> >     |          **Our BEACON-B**         |                |
> >     |              BEACON-B             |   26.01±1.81   |
> >     |            BEACON-B512            | **28.17±1.81** |
> >
> > [r16] Labun, K., Montague, T. G., Gagnon, J. A., Thyme, S. B., & Valen, E. (2016). CHOPCHOP v2: a web tool for the next generation of CRISPR genome engineering. Nucleic acids research, 44(W1), W272-W276.
> >
> > [r17] Chuai, G., Ma, H., Yan, J., Chen, M., Hong, N., Xue, D., ... & Liu, Q. (2018). DeepCRISPR: optimized CRISPR guide RNA design by deep learning. Genome biology, 19, 1-18.
> >
> > [r18] Wong, N., Liu, W., & Wang, X. (2015). WU-CRISPR: characteristics of functional guide RNAs for the CRISPR/Cas9 system. Genome biology, 16, 1-8.
> >
> > [r19] Chari, R., Yeo, N. C., Chavez, A., & Church, G. M. (2017). sgRNA Scorer 2.0: a species-independent model to predict CRISPR/Cas9 activity. ACS synthetic biology, 6(5), 902-904.
> >
> > [r20] Prykhozhij, S. V., Rajan, V., Gaston, D., & Berman, J. N. (2015). CRISPR multitargeter: a web tool to find common and unique CRISPR single guide RNA targets in a set of similar sequences. PloS one, 10(3), e0119372.
> >
> > [r21] Heigwer, F., Kerr, G., & Boutros, M. (2014). E-CRISP: fast CRISPR target site identification. Nature methods, 11(2), 122-123.
> >
> > [r22] Doench, J. G., Fusi, N., Sullender, M., Hegde, M., Vaimberg, E. W., Donovan, K. F., ... & Root, D. E. (2016). Optimized sgRNA design to maximize activity and minimize off-target effects of CRISPR-Cas9. Nature biotechnology, 34(2), 184-191.
> >
> > - **CRISPR Off-Target Prediction**
> >
> >     |               Method              |  Metric (SC %) |
> >     |:---------------------------------:|:--------------:|
> >     |         **Literature SOTA**        |                |
> >     |                CFD [r23]               |      10.3      |
> >     |                MIT [r24]               |       8.0      |
> >     |               CCTop  [r25]             |       5.2      |
> >     |              CROP-IT  [r26]            |       4.2      |
> >     |             DeepCRISPR            |    **12.6**    |
> >     |     **Naive Supervised Model**    |                |
> >     |                CNN                |   11.40±0.10   |
> >     |               ResNet              | **11.50±0.22** |
> >     |                LSTM               |    8.60±0.13   |
> >     | **Pretrained RNA Language Model** |                |
> >     |               RNA-FM              |    2.49±1.56   |
> >     |              RNABERT              |    4.27±1.05   |
> >     |              RNA-MSM              |    3.85±0.99   |
> >     |        SpliceBERT-H510        |    4.00±1.13   |
> >     |          SpliceBERT-MS510         |    3.49±2.12   |
> >     |         SpliceBERT-MS1024         |    **5.00±0.71**   |
> >     |             UTR-LM-MRL            |    4.28±0.15   |
> >     |            UTR-LM-TE&EL           |    2.91±1.18   |
> >     |            UTRBERT-3mer           |    4.48±1.12   |
> >     |            UTRBERT-4mer           |    3.11±1.10   |
> >     |            UTRBERT-5mer           |    3.93±0.24   |
> >     |            UTRBERT-6mer           |    4.90±0.57   |
> >     |          **Our BEACON-B**         |                |
> >     |              BEACON-B             |    **4.42±0.33**   |
> >     |            BEACON-B512            |    3.82±1.04   |
> >
> > [r23] Doench, J. G., Fusi, N., Sullender, M., Hegde, M., Vaimberg, E. W., Donovan, K. F., ... & Root, D. E. (2016). Optimized sgRNA design to maximize activity and minimize off-target effects of CRISPR-Cas9. Nature biotechnology, 34(2), 184-191.
> >
> > [r24] Hsu, P. D., Scott, D. A., Weinstein, J. A., Ran, F. A., Konermann, S., Agarwala, V., ... & Zhang, F. (2013). DNA targeting specificity of RNA-guided Cas9 nucleases. Nature biotechnology, 31(9), 827-832.
> >
> > [r25] Stemmer, M., Thumberger, T., del Sol Keyer, M., Wittbrodt, J., & Mateo, J. L. (2015). CCTop: an intuitive, flexible and reliable CRISPR/Cas9 target prediction tool. PloS one, 10(4), e0124633.
> >
> > [r26] Singh, R., Kuscu, C., Quinlan, A., Qi, Y., & Adli, M. (2015). Cas9-chromatin binding information enables more accurate CRISPR off-target prediction. Nucleic acids research, 43(18), e118-e118.

---

> > ### Author Rebuttal · Authors · 2024-08-17
> >
> > ## O2: Justifications for Prevention of Information Leakage
> >
> > - For downstream tasks, we followed established, peer-reviewed data processing pipelines for tasks like contact map prediction and APA isoform prediction (see details in `O4: Detailed Preprocessing for Each Task` below). Additionally, after preprocessing, we double-check to ensure there is no overlap between the training and testing datasets.
> > - For the unsupervised pre-training, since the model is trained on sequences without corresponding labels, we do not need to worry about data overlap between the pre-training and downstream tasks.
> >
> > ## O3: Detailed Ablation Studies for BEACON-B
> >
> > - In fact, we have analyzed the components of RNA language models in Section 5.4 of our paper, specifically focusing on tokenizers and positional encoding in a from-scratch training setting, as detailed in Table 4. This analysis led to the design of BEACON-B, which integrates Single Tokenization and ALiBi positional encoding based on insights from the ablation studies.
> >
> > - Furthermore, we carried out detailed ablation experiments in the pre-training setting. We used the same pre-training setup of BEACON-B512 (pre-training on 512-length sequences of human-filtered ncRNA and using FlashAttn). This setup allowed us to scrutinize the impact of various tokenizers and positional encodings. For downstream evaluation, in order to minimize computational resources and accelerate experimental procedures, we evaluated three major RNA task categories: Distance Map Prediction (DMP) from structural tasks, APA Isoform Prediction (APA) from functional tasks, and Vaccine Degradation Prediction (VDP) from engineering tasks.
> >
> >
> >     |             |    |  | CMP   | APA   | VDP   |
> >     |-------------------|-------------|---------------------|-------|-------|-------|
> >     |   Method       |  Tokenizer  |  Positional Encoding  | Metric (P@L %) | Metric ($R^2$ %) | Metric (MCRMSE↓)  |
> >     | BPE-ALiBi         | BPE         | ALiBi               | 49.08 | 64.98 | 0.640  |
> >     | Non-overlap-ALiBi | Non-overlap | ALiBi               | 47.68 | 66.61 | 0.639 |
> >     | 6mer-ALiBi        | 6mer        | ALiBi               | 48.79 | 70.77 | 0.372 |
> >     | Single-RoPE       | Single      | RoPE                | 12.84 | 33.86 | 0.462 |
> >     | Single-APE        | Single      | APE                 | 50.76 | 66.20  | 0.327 |
> >     | BEACON-B512       | Single      | ALiBi               | **56.82** | **72.00**    | **0.320**  |
> >
> >     The results of these ablation experiments further underscored that BEACON-B's design, combining Single Tokenization with ALiBi positional encoding, remains the most efficacious approach.
> >
> > ## O4: Detailed Preprocessing for Each Task
> >
> > The processing of data for each task is discussed in detail below. We will include them in the appendix of the revised version.
> > - **Non-coding RNA Function Classification**
> >
> >      We followed nRC [r15] and sourced non-coding RNA sequences from the Rfam database, recognized for its comprehensive collection of manually curated RNA sequences. We specifically selected 13 diverse ncRNA classes including miRNA, 5S rRNA, 5.8S rRNA, ribozymes, CD-box, HACA-box, scaRNA, tRNA, Intron gpI, Intron gpII, IRES, leader and riboswitch. Utilizing the CD-HIT tool[r27], we reduced sequence redundancy to 20%, ensuring a representative yet manageable dataset size. This process allowed us to include a wide array of RNA types while controlling data complexity.
> >
> >     For each selected class, we systematically assembled a training-validation dataset comprising 500 sequences per class, except for the IRES class where only 320 sequences were available, leading to a total of 6320 ncRNA sequences. We subsequently divided it into two groups: a validation set comprising 650 sequences, with 50 from each class, and a training set consisting of the remaining 5670 sequences. The balancing was further refined in our test dataset, which contains an equal number of sequences from each class, totaling 2600 sequences, ensuring that our model evaluation would not be biased by uneven class representation.
> >
> >     [r27] Fu, L., Niu, B., Zhu, Z., Wu, S., & Li, W. (2012). CD-HIT: accelerated for clustering the next-generation sequencing data. Bioinformatics, 28(23), 3150-3152.
> >
> > - **Secondary Structure Prediction**
> >
> >     We followed the preprocessing from the bpRNA-1m dataset [r28]. To mitigate sequence redundancy and enhance the diversity of our dataset, we applied an 80% sequence-identity cut-off and restricted the maximum sequence length to below 500 nucleotides, similar to the process described in both referenced articles. This step was crucial in reducing the risk of overfitting and ensuring that our models are trained on genetically distinct samples.
> >
> >     The dataset was strategically split into three parts: a training set (TR0), a validation set (VL0), and a test set (TS0). This separation was carried out randomly to avoid any biases that might affect the evaluation of the model’s performance.
> >
> >     [r28] Danaee, P., Rouches, M., Wiley, M., Deng, D., Huang, L., & Hendrix, D. (2018). bpRNA: large-scale automated annotation and analysis of RNA secondary structure. Nucleic acids research, 46(11), 5381-5394.

---

> > ### Author Rebuttal · Authors · 2024-08-17
> >
> > - **Contact Map Prediction**
> >
> >     We choose the benchmark datasets used by [r13, r29], which are constructed based on a set of nonredundant RNA 3D structures from Leontis and Zirbel (2012) (Version 3.99, 2019-11-06), containing 1786 entries with resolution < 4  ̊A initially. Sequences with length < 32nt or > 1000nt or with redundancy over 80% as well as with too few positive points (< 5) are removed. Finally, 221 sequences left are used for training and 80 sequences for testing. Using their pdb files, We then compute the ground truth pairwise tertiary distance, which is defined as the minimal atomic distance of the two bases. Then the pairwise contact is defined as the distance between two bases is less than 8Å.
> >
> >     [r29] Chen, J., Hu, Z., Sun, S., Tan, Q., Wang, Y., Yu, Q., ... & Li, Y. (2022). Interpretable RNA foundation model from unannotated data for highly accurate RNA structure and function predictions. arXiv preprint arXiv:2204.00300.
> >
> > - **Distance Map Prediction**
> >
> >     The dataset used for distance map prediction is the same as the contact map prediction. We first compute the pairwise tertiary distance as above. Then we limit the distance from 0 to 20 ̊A and regard the value over 20 as 20. Finally, we use 20 to normalize the distance values and obtain a normalized distance map with elements falling into [0, 1].
> >
> > - **Structural Score Imputation**
> >
> >     We followed the preprocessing procedure as outlined in [r30]. The icSHAPE HEK293 cell line raw sequencing data (~300 million clean reads) was downloaded and processed using icSHAPE-pipe to obtain structure profiles for 4,091 transcripts. The full-length transcript structure profiles were then binned into non-overlapping 100-nt fragments. 21,859 fragments were retained with valid structural scores for every nucleotide.
> >
> >     These valid fragments were randomly split into training and test sets in a 7:3 ratio, resulting in 15,085 fragments for training and 6,774 for testing. 569 fragments, with high sequence similarity to any fragment in the training set based on BLAST results, were removed from the test set.
> >
> >     For the training set, 30% of the nucleotides were randomly masked as null, with the ground-truth structural scores of these positions used as training labels. For the test set, the original icSHAPE dataset was downsampled to 100 million clean reads, with the structure profiles to simulate lower sequencing depth recalculated. This process generated missing values, and only fragments with at least one valid structural score were retained, resulting in 3,095 fragments in the final test dataset.
> >
> >     [r30] Gong, J., Xu, K., Ma, Z., Lu, Z. J., & Zhang, Q. C. (2021). A deep learning method for recovering missing signals in transcriptome-wide RNA structure profiles from probing experiments. Nature Machine Intelligence, 3(11), 995-1006.
> >
> > - **Splice Site Prediction**
> >
> >     We followed the preprocessing protocol from [r31]. The GENCODE V24lift37 gene annotation table was downloaded from the UCSC table browser, and 20,287 protein-coding gene annotations were extracted. For genes with multiple isoforms, the principal transcript was selected. Genes lacking splice junctions were removed. The remaining genes were split into training and test sets: chromosomes 2, 4, 6, 8, 10-22, X, and Y were used for training, with 1/9 reserved for early stopping. For testing, genes from chromosomes 1, 3, 5, 7, and 9 without paralogs were selected.
> >
> >     For each gene, mRNA transcript sequences from the canonical transcription start to end sites were extracted using the hg19/GRCh37 assembly. Sequences were zero-padded to a multiple of 5,000 nucleotides and split into blocks of length 5,000. Then, the center 100 nucleotides of each block were extracted as the input sequence, and the corresponding splice site label was extracted as the output. (non-splice sites [1, 0, 0], splice acceptors [0, 1, 0], and splice donors [0, 0, 1])
> >
> >     [r31] Jaganathan, K., Panagiotopoulou, S. K., McRae, J. F., Darbandi, S. F., Knowles, D., Li, Y. I., ... & Farh, K. K. H. (2019). Predicting splicing from primary sequence with deep learning. Cell, 176(3), 535-548.

---

> > ### Author Rebuttal · Authors · 2024-08-17
> >
> > - **APA Isoform Prediction**
> >
> >     The preprocessing for IPA Isoform began with filtering the raw sequencing reads from all MPRAs [r32] to ensure high-quality, full-length RNA reads. These reads were clustered based on the randomized region upstream of the proximal polyadenylation site (pPAS), creating a dictionary of sequence variants for each library. To expand the dictionary, the plasmid library was sequenced to include members that did not express a distal isoform, and RNA reads were mapped to their respective dictionary entries by matching the upstream region with the shortest Hamming distance.
> >
> >     For each mapped read, the polyadenylation cleavage site was identified by scanning for the Poly-A tail. The cleavage positions were stored as vectors associated with each sequence variant, including a special position for reads mapping to non-random distal sites. The resulting dataset consisted of a dictionary of unique sequence variants with associated cleavage-position count vectors, which then underwent a final filtering step. This step selected high-confidence variants by removing sequences supported by fewer than 10-20 unique UMI RNA reads or sequences with over 75% A-nucleotides in a 12-20 bp region to avoid internal priming artifacts.
> >
> >     Data from 12 random 3' UTR libraries were processed, with 9 used for training and 3 held out (the 3 held-out libraries were not used in the current analysis). To ensure balanced representation in the test set, the sequences from each library were individually shuffled based on the reada count, then combined using a round-robin order, placing one sequence from each library after another in descending order of read count. This process ensured that the test set contained an equal number of high-read count sequences from each library. The remaining sequences were placed at the beginning of the combined library, and the training set was further shuffled. This approach ensured a balanced, high-quality dataset for training, validation, and testing. To maintain balanced and high-quality data for benchmarking, the top 10% of high-read count sequences were selected, and the most highly expressed sequences were further chosen for testing.
> >
> >     [r32] Shigaki, D., Adato, O., Adhikari, A. N., Dong, S., Hawkins‐Hooker, A., Inoue, F., ... & Beer, M. A. (2019). Integration of multiple epigenomic marks improves prediction of variant impact in saturation mutagenesis reporter assay. Human mutation, 40(9), 1280-1291.
> >
> > - **Modification Prediction**
> >
> >     We followed MultiRM [r33] and compiled a comprehensive dataset comprising 20 epi-transcriptome profiles derived from 15 different base-resolution technologies, addressing 12 distinct RNA modifications such as m6A, m1A, m5C, among others. A key focus was given to constructing a reliable set of negative control data, which was crucial for our predictors' accuracy. To achieve this, negative sites (non-modified nucleotides) were carefully selected from the unmodified bases within the same transcripts that contained the positive modification sites, ensuring an authentic comparison baseline.
> >
> >     Moreover, the dataset was divided into three distinct subsets: training, validation, and testing. The training set was purposefully left unbalanced to mirror the natural prevalence differences among the RNA modification types, which introduces realistic challenges in model training akin to real-world scenarios. Conversely, the validation and test sets were meticulously balanced—each set containing 150 and 50 samples, respectively, across all modification types—to ensure fairness in model evaluation and performance metrics.
> >
> >     [r33] Song, Z., Huang, D., Song, B., Chen, K., Song, Y., Liu, G., ... & Meng, J. (2021). Attention-based multi-label neural networks for integrated prediction and interpretation of twelve widely occurring RNA modifications. Nature communications, 12(1), 4011.
> >
> > - **Mean Ribosome Loading**
> >
> >     For 5'UTR processing, we primarily employed a large-scale synthetic Human 5'UTR library [r34], which comprised 83,919 sequences of varying lengths, as articulated in the first referenced article. The validation set was meticulously crafted by evenly sampling 7600 sequences across these different lengths to ensure fair representation and robust generalizability testing. The remaining sequences were utilized for training purposes. To further enhance our model’s performance assessment, we incorporated an additional dataset of 7600 real human 5’UTRs, mirroring the length distribution provided by the synthetic library. This dual-dataset strategy not only allowed for a thorough evaluation of the model's predictive accuracy but also its ability to generalize across synthetic and real-world data.
> >
> >     Additionally, as described in the second referenced article, the pivotal role of the 5’UTR sequence in translation efficiency was explored through rigorous experiments utilizing data from Massively Parallel Reporter Assays (MPRAs), which included a substantial library of 280,000 gene sequences. This extensive dataset facilitated the fine-tuning of our model, Uni-RNA, to predict Mean Ribosome Load (MRL) effectively.
> >
> >     [r34] Sample, P. J., Wang, B., Reid, D. W., Presnyak, V., McFadyen, I. J., Morris, D. R., & Seelig, G. (2019). Human 5′ UTR design and variant effect prediction from a massively parallel translation assay. Nature biotechnology, 37(7), 803-809.

---

> > ### Author Rebuttal · Authors · 2024-08-17
> >
> > - **Vaccine Degradation Prediction**
> >
> >     We followed OpenVaccine [r35] to collect 2400 sequences specifically for training purposes, 629 sequences were made available for public testing during the competition, and the remainder were reserved for private scoring. Post-competition, all these sequences are now available with comprehensive labels detailing various degradation rates under multiple experimental conditions, such as high pH and high temperature, both with and without magnesium.
> >
> >     To ensure the integrity of the public leaderboard and prevent biases, we implemented rigorous filtering criteria on the 629 sequences used for public testing. These criteria included setting a minimum value threshold across all experimental conditions and ensuring a mean signal-to-noise ratio above a specified level.
> >
> >     We also provide the test data of the original private leaderboard although we do not use it in the paper. For the private leaderboard data, which involves the most critical evaluation, we included measurements from an additional 3005 RNA sequences, which are slightly longer. These were processed to include data for the first 91 bases, carefully excluding the final bases to align with experimental limitations.
> >
> >
> >     [r35] Wayment-Steele, H. K., Kladwang, W., Watkins, A. M., Kim, D. S., Tunguz, B., Reade, W., ... & Das, R. (2022). Deep learning models for predicting RNA degradation via dual crowdsourcing. Nature Machine Intelligence, 4(12), 1174-1184.
> >
> >
> > - **Programmable RNA Switches**
> >
> >     We followed the data generation pipeline from [r36]. A toehold-switch library based on 244,000 putative trigger sequences was designed and synthesized, which covered the complete genomes of 23 pathogenic viruses, the entire coding regions of 906 human transcription factors, and approximately 10,000 random sequences. The synthesized oligo pool was used to generate two construct libraries for ON and OFF states, which were transformed into BL21 *E. coli*. The OFF library contained toehold-switch constructs without triggers, while the ON library contained the same toeholds with complementary triggers fused to their corresponding switches.
> >
> >     These libraries were sorted using fluorescence-activated cell sorting (FACS) into four bins, and the variants in each bin were quantified using next-generation sequencing (NGS) to determine their fluorescence distributions. After quality control, the toehold-switch library included 109,067 ON-state measurements, 163,967 OFF-state measurements, and 91,534 ON/OFF paired ratios, where both states were characterized for a given switch. The ON and OFF data were normalized from 0 to 1, resulting in ON/OFF ratios normalized from -1 to 1. Following [r36], a quality control process was applied to remove artifacts and ensure the reliability of the data. There are 5 levels of quality control (QC1, QC2, QC3, QC4, and QC5), with QC1 being the lowest quality and QC5 being the highest. All datasets at QC levels above QC2 are used for training with Q5 left for testing.
> >
> >
> >     [r36] Angenent-Mari, N. M., Garruss, A. S., Soenksen, L. R., Church, G., & Collins, J. J. (2020). A deep learning approach to programmable RNA switches. Nature communications, 11(1), 5057.
> >
> > - **CRISPR On-Target Prediction**
> >     For the processing of the on-target sgRNA dataset, we sourced experimentally validated sgRNAs targeting approximately 1,071 genes across four distinct cell lines[r17], namely hct116, hek293t, hela, and hl60. To ensure the integrity of our dataset, we selected hl60 cell line data and removed redundant entries and restricted our dataset to sgRNAs with direct experimental validation of knockout efficacy, quantified as the log-fold change.
> >
> >     Furthermore, for a balanced and normalized dataset conducive to regression analysis, we employed a collaborative filtering-based normalization approach [r17], akin to methodologies used in user-item recommendation systems. We constructed an efficacy matrix where rows represented experiments and columns represented sgRNAs. The knockout efficacy was normalized by calculating the mean values across the rows, columns, and the entire matrix, and then adjusting the sgRNA efficacy scores by subtracting these mean values and dividing by the number of mean types considered. This normalization process ensures that our dataset remains unbiased and reflective of true knockout efficacies across various experimental setups, thereby enhancing the generalizability and accuracy of our predictive models.
> >
> > - **CRISPR Off-Target Prediction**
> >
> >     We followed the dataset from DeepCRISPR  [r17] including off-target profiles from two distinct cell types: 293-related cell lines and K562 cells, encompassing 30 sgRNAs in total. Using the bowtie2 tool [r37], we used K562 cells data and identified approximately 160,000 potential off-target loci across the genome, allowing for up to six nucleotide mismatches per sgRNA. This resulted in a highly unbalanced dataset, with varying numbers of loci associated with each level of mismatch, from one to six.
> >
> >     To address the imbalance and refine our dataset for the regression models, we labeled the off-target sites and normalized them according to the indel frequency detected by various genome-wide off-target detection techniques. This normalization process helped to mitigate the skewness introduced by the uneven distribution of loci, ensuring that our model could generalize effectively across different genomic backgrounds and detection assays.
> >
> >
> >     [r37] Langmead, B., & Salzberg, S. L. (2012). Fast gapped-read alignment with Bowtie 2. Nature methods, 9(4), 357-359.

---

> > ### Author Rebuttal · Authors · 2024-08-17
> >
> > ## O5: Computational Complexity Comparisons
> >
> > - We have compared the currently available pre-training resource consumption of RNA language models in Table 5 in Section 5.5, shown via GPU Days (Days * GPUs).
> >
> > - We also compared the FLOPs and MACs of the different models. The detailed computational costs of the models are outlined as follows (following the sequence lengths at which the models were pretrained). Despite our model employing a vanilla BERT-base model with 12 layers, which doesn’t minimize FLOPs, it still manages to achieve the lowest resource usage, requiring only 3.58 GPU Days. This efficiency is primarily due to our use of a modestly sized, filtered ncRNA dataset combined with a lightweight pre-training schedule (80K steps and a batch size of 512).
> >
> >     |       Method      | FLOPs (G) | MACs (G) | GPU Days  (Days * GPUs) |
> >     |:-----------------:|:---------:|:--------:|:--------:|
> >     |       RNA-FM      |   233.88  |  116.77  | 30 * 8=240 |
> >     |      RNABERT      |    0.91   |   0.46   | - |
> >     |      RNA-MSM      |   201.46  |   84.56  | - |
> >     |  SpliceBERT-H510  |   19.27   |   9.63   | 7 * 8=56 |
> >     |  SpliceBERT-MS510 |   19.27   |   9.63   | 7 * 8=56 |
> >     | SpliceBERT-MS1024 |   38.69   |   19.33  | 7 * 8 +3 * 4=68 |
> >     |     UTR-LM-MRL    |    5.76   |   2.83   | - |
> >     |    UTR-LM-TE&EL   |    5.76   |   2.83   | - |
> >     |    UTRBERT-3mer   |   511.41  |  255.56  | 38 * 4=152 |
> >     |    UTRBERT-4mer   |   511.41  |  255.56  | 38 * 4=152 |
> >     |    UTRBERT-5mer   |   511.41  |  255.56  | 38 * 4=152 |
> >     |    UTRBERT-6mer   |   511.41  |  255.56  | 38 * 4=152 |
> >     |      BEACON-B     |   198.48  |   99.13  | 1.3 * 8=**10.4** |
> >     |    BEACON-B512    |   87.02   |   43.49  | 0.895 * 4=**3.58** |

---

> > > ### Comment · Reviewer_Wr93 · 2024-08-29
> > > **Response to rebuttal**
> > >
> > > I appreciate the authors' detailed explanations and nice work in adding additional experiments. With these efforts my concerns are basically addressed. I have raised my score.

---

> > > > ### Author Response · Authors · 2024-08-29
> > > > **Thanks for the response**
> > > >
> > > > Thank you so much for your support and for raising your score to 6. We are pleased that the explanations and additional experiments addressed your concerns, and we will include all these detailed descriptions and experiments in the final version. If you have any further suggestions or issues, please don't hesitate to let us know. We are always open to feedback and committed to improving the paper.

---

### Official Review · Reviewer_M1uB · 2024-07-27
**Review of BEACON**

**Rating:** 8
**Confidence:** 4

**Review:**

Pros 1: The paper starts off with a comprehensive scope of the field of deep learning for RNA-related applications and precipitates these applications in 13 tasks that can be used to evaluate any models in the field. BEACON is a well-rounded benchmark, evaluated using both traditional and advanced RNA language models and appropriate metrics for different tasks.

Pros 2: The paper is well-structured and provides sufficient context and delineation for each of the 13 different tasks.

Cons 1: The paper does not properly document the implementation of these models or the baseline model, making it difficult for researchers to reproduce the results.

Cons 2: The paper does not take inverse RNA folding models into consideration.

Cons 3: The paper does not take model specialization into consideration. Language models may be designed for highly specific tasks and may not perform as well in a broad benchmark. The paper does not describe a way to prevent such models from being unfairly penalized in the rankings.

**Strengths:**

This submission is methodologically rigorous, and the introduction of BEACON-B makes the baseline more practical and accessible. Today, RNA is crucial to research on genetic regulation and disease mechanisms, making an advance in RNA research hold significant benefits to the scientific community.

**Additional Feedback:**

NA

**Clarity:**

While the paper discusses in great detail how BEACON evaluates previously published models in the niche, the paper does not talk about any other molecular biology benchmarks, such as MoleculeNet or TAPE for protein sequencing. This contextualizing comparison could highlight how BEACON extends or deviates from methodologies used in benchmarking for molecular biology.

**Correctness:**

The paper makes well-supported claims and the core contribution of the paper, that is, the BEACON benchmark consists of a wide array of tasks that are then evaluated using standard practices in DL benchmarking. The paper tailors specific metrics to the specifications of each RNA task, such as F1 score and AUC for classification tasks and  R² and MCRMSE for regression tasks. By including multiple baselines and state-of-the-art models, the authors provide a thorough comparison framework.

**Documentation:**

While BEACON provides a strong, new baseline for RNA-related tasks in DL, it could use additional details that enhance reproducibility. For example, more detailed information on the specific configurations, hyperparameters, and training settings used for each model would help replicate the experiments precisely. Similarly, a detailed description of preprocessing steps, such as any normalization, filtering, or augmentation should be documented. Data splits should be discussed in more detail.

**Limitations:**

There are no negative social impacts on the development of BEACON.

**Opportunities For Improvement:**

The paper does not take model specialization into consideration. Language models may be designed for highly specific tasks and may not perform as well in a broad benchmark. The paper does not describe a way to prevent such models from being unfairly penalized in the rankings.
The paper does not take inverse RNA folding models into consideration, which is quickly becoming a significant area of research.
The tasks are pertinent but arbitrary and lack real-world validation. The tasks in the paper, especially the engineering applications subsection, assume practical relevance to the current RNA research scenario but make it crucial for the baseline model to be updated frequently to include the nuances and new applications in the field. Although diverse, the benchmark may not encompass all RNA-related tasks.
In the paper,  aggregating performance metrics across diverse tasks can lead to a loss of information on specific strengths and weaknesses, making the overall ranking less informative.

**Relation To Prior Work:**

While the paper discusses in great detail how BEACON evaluates previously published models in the niche, the paper does not talk about any other molecular biology benchmarks, such as MoleculeNet or TAPE for protein sequencing. This contextualizing comparison could highlight how BEACON extends or deviates from methodologies used in benchmarking for molecular biology.

**Summary And Contributions:**

This paper introduces BEACON, a benchmarking method developed to address a lack of standardization in the field of RNA models. BEACON tests models on 13 RNA-related taks that cover sequence- and nucleotide-level analyses as performed by previously available models in the fields of (1) structural analysis, (2) functional analysis, and (3) de-novo engineering applications.

The authors select from a diverse range of previously published models, including DL models such as CNNs, ResNets, and LSTMs, as well as advanced RNA foundation models like RNA-FM, UTR-LM, RNA-BERT, RNA-MSM, SpliceBERT, and 3UTRBERT. When ev aluated against each other, pre-trained RNA language models surpassed state-of-the-art models on many tasks. To perform this evaluation, the authors used traditional metrics. For classification tasks, they used  F1 score, AUC, Top K Accuracy, and Top L Precision, whereas for regression tasks, metrics such as R², MCRMSE, and Spearman Correlation.

The key contributions made by this paper are as follows:
BEACON is the first benchmark for RNA-related tasks in deep learning.
The paper evaluated several pre-trained RNA language models against each other.

---

> ### Author Rebuttal · Authors · 2024-08-17
>
> We sincerely appreciate Reviewer M1uB's thoughtful feedback, and here we provide corresponding responses to address these concerns.
>
> ## O1: Document for The Implementation
>
> In parallel with this manuscript, we also develop a library named [MultiMolecule](https://github.com/DLS5-Omics/multimolecule/tree/7554a94) accompanied by the [BEACON codebase](https://github.com/terry-r123/RNABenchmark). All implementation details of compared models can be found in the `Model Details` and `Training Details` sections of each model on the [document website](https://multimolecule.danling.org/models/). Additionally, the converted pretrained weights are publicly available on the [Huggingface Hub](https://huggingface.co/multimolecule), facilitating reproducibility and further research. Besides, implementation details of our baseline models will be included in the revised version.
>
> ## O2: Consideration for Inverse RNA Folding
>
> Thanks for the insightful suggestion. We genuinely agree that inverse RNA folding is a critical task in the field of synthetic RNA research. Overall, this task is not included in the current version mainly due to the following reasons:
>
> - Our benchmark focuses on RNA language models, and most RNA language models are not specifically designed for inverse tasks. The primary goal of current RNA language models is to understand sequence-related analytical tasks, with inverse folding not being the main objective. Technically, current RNA language models are trained to input sequences and do not yet accept structures as inputs, which is why we did not include inverse RNA folding in our benchmark.
>
> - However, although the task of inverse RNA folding extends beyond the current scope of this benchmark, we plan to extend our benchmark to support more complex tasks involving diverse types of input, broadening our coverage in RNA research significantly in the future. Specifically, for inverse RNA folding, we will consider datasets and tasks such as Eterna100v1 [r1] and Eterna100v2 [r2] for standard inverse RNA folding evaluations and PseudoBase++ [r3] to evaluate the capability of inverse folding extending to pseudoknots. Regarding models, we will evaluate significant inverse RNA folding methods such as RNAinverse [r4], MCTS-RNA [r5], LEARNA [r6], MetaLEARNA [r6], gRNAde [r7], and RiboDiffusion [r8]. We are also considering influential inverse protein folding models like StructGNN [r9], GVP-GNN [r10], and ProteinMPNN [r11], PiFold [r12], given their potential to offer new insights into RNA structure modeling, owing to the similarities between modeling protein and RNA structures. For evaluation metrics, we could consider assessing sequence similarity, sequence diversity, and structural similarity, using indicators such as recovery rate, diversity, and structural similarity.
>
> [r1] Anderson-Lee, J., Fisker, E., Kosaraju, V., Wu, M., Kong, J., Lee, J., ... & Players, E. (2016). Principles for predicting RNA secondary structure design difficulty. Journal of molecular biology, 428(5), 748-757.
>
> [r2] Koodli, R. V., Rudolfs, B., Wayment-Steele, H. K., Eterna Structure Designers, & Das, R. (2021). Redesigning the EteRNA100 for the Vienna 2 folding engine. BioRxiv, 2021-08.
>
> [r3] Van Batenburg, F. H. D., Gultyaev, A. P., Pleij, C. W. A., Ng, J., & Oliehoek, J. (2000). PseudoBase: a database with RNA pseudoknots. Nucleic Acids Research, 28(1), 201-204.
>
> [r4] Hofacker, I. L., Fontana, W., Stadler, P. F., Bonhoeffer, L. S., Tacker, M., & Schuster, P. (1994). Fast folding and comparison of RNA secondary structures. Monatshefte fur chemie, 125, 167-167.
>
> [r5] Yang, X., Yoshizoe, K., Taneda, A., & Tsuda, K. (2017). RNA inverse folding using Monte Carlo tree search. BMC bioinformatics, 18, 1-12.
>
> [r6] Runge, F., Stoll, D., Falkner, S., & Hutter, F. (2018). Learning to design RNA. arXiv preprint arXiv:1812.11951.
>
> [r7] Joshi, C. K., Jamasb, A. R., Viñas, R., Harris, C., Mathis, S. V., Morehead, A., ... & Liò, P. (2024). gRNAde: Geometric Deep Learning for 3D RNA inverse design. bioRxiv.
>
> [r8] Huang, H., Lin, Z., He, D., Hong, L., & Li, Y. (2024). RiboDiffusion: Tertiary Structure-based RNA Inverse Folding with Generative Diffusion Models. bioRxiv, 2024-04.
>
> [r9] Ingraham, J., Garg, V., Barzilay, R., & Jaakkola, T. (2019). Generative models for graph-based protein design. Advances in neural information processing systems, 32.
>
> [r10] Jing, B., Eismann, S., Suriana, P., Townshend, R. J. L., & Dror, R. (2020, September). Learning from protein structure with geometric vector perceptrons. In International Conference on Learning Representations.
>
> [r11] Dauparas, J., Anishchenko, I., Bennett, N., Bai, H., Ragotte, R. J., Milles, L. F., ... & Baker, D. (2022). Robust deep learning–based protein sequence design using ProteinMPNN. Science, 378(6615), 49-56.
>
> [r12] Gao, Z., Tan, C., Chacón, P., & Li, S. Z. (2022). PiFold: Toward effective and efficient protein inverse folding. arXiv preprint arXiv:2209.12643.

---

> > ### Author Rebuttal · Authors · 2024-08-17
> >
> > ## O3: More Real-World Applications
> >
> > Thanks for this comment. We focus on RNA language models for the first comprehensive benchmark, and the absence of some important tasks is due primarily to two reasons:
> >
> > - We standardize input formats by using RNA sequences across all tasks and all models. This approach was necessary to maintain consistency in evaluations. However, it may not fully capture the potential of models that are designed to leverage richer input data, such as the scRNA-seq data for cell annotation and gene regulation network which uses genes with expression level [r13] and inverse RNA folding using structure as input [r14]. Future work could explore alternative evaluation settings that allow for the use of more complex inputs, thereby providing a more nuanced assessment of model performance （specific examples are mentioned in `O2: Consideration for Inverse RNA Folding` .
> >
> > - While our current benchmark effectively evaluates RNA language models across a spectrum of tasks, we acknowledge the out-of-selection of certain complex tasks such as 3D RNA structure prediction. This is primarily due to several challenges: 1) The majority of open-source RNA language models have not been developed to perform 3D structure prediction tasks. 2) The acquisition of high-quality structural data, often requiring Multiple Sequence Alignment (MSA) data, is prohibitively expensive. This, coupled with the limited availability of such data, poses a significant barrier [r15]. 3) The computational pipeline for predicting 3D structures (e.g., structural modules) is exceedingly complex and resource-intensive, often necessitating days of computation on high-performance GPUs [r16].
> >     In future extensions of our benchmark, we aim to incorporate more diverse tasks, including those requiring complex data inputs and significant computational resources, to better encompass the breadth of challenges in RNA research.
> >
> > [r13] Theodoris, C. V., Xiao, L., Chopra, A., Chaffin, M. D., Al Sayed, Z. R., Hill, M. C., ... & Ellinor, P. T. (2023). Transfer learning enables predictions in network biology. Nature, 618(7965), 616-624.
> >
> > [r14] Hofacker, I. L., Fontana, W., Stadler, P. F., Bonhoeffer, L. S., Tacker, M., & Schuster, P. (1994). Fast folding and comparison of RNA secondary structures. Monatshefte fur chemie, 125, 167-167.
> >
> > [r15] Wang, W., Feng, C., Han, R., Wang, Z., Ye, L., Du, Z., ... & Yang, J. (2023). trRosettaRNA: automated prediction of RNA 3D structure with transformer network. Nature Communications, 14(1), 7266.
> >
> > [r16] Li, Y., Zhang, C., Feng, C., Pearce, R., Lydia Freddolino, P., & Zhang, Y. (2023). Integrating end-to-end learning with deep geometrical potentials for ab initio RNA structure prediction. Nature Communications, 14(1), 5745.
> >
> > ## O4:  Consideration for Model Specialization
> >
> > We appreciate the reviewer's concerns regarding model specialization and the potential for unfair penalization in overall rankings. We clarify that while the primary objective of this manuscript is to establish a comprehensive benchmark, it does not aim to produce a definitive ranking of models. Instead, our focus is on showcasing the distinct characteristics and strengths of each model. To address specific strengths, we have included task-specific evaluations in Table 3, where the top 4 performing models for each task are highlighted. This method ensures that models with particular specializations are recognized for their capabilities in those areas, rather than being penalized in a broader comparison. Through this approach, we aim to provide researchers with clear insights into which models perform best under specific conditions and to highlight any potential underlying connections between tasks that emerge from our comparative analyses.

---

### Official Review · Reviewer_vm5g · 2024-07-31
**Important RNA benchmark**

**Rating:** 8
**Confidence:** 3
**Correctness:** yes
**Clarity:** yes

**Review:**

The paper is well written and well motivated. The tasks are well explained and relevant to the task at hand. Furthermore the baselines are comprehensive and provide interesting insights such as LSTMS and CNN being still competitive compared to LLMs. I think this will provide a good benchmark to push the field forward.

**Strengths:**

See above

**Additional Feedback:**

none

**Documentation:**

yes

**Limitations:**

yes

**Opportunities For Improvement:**

Explain more the limitations highlighted in the conclusion. What aspects of Biology might be missing from this benchmark?

**Relation To Prior Work:**

yes

**Summary And Contributions:**

The paper presents a benchmark focused on RNA tasks and models:

- Identifies 3 broad categories of tasks: structural, functional, and engineering applications
- Compiles public datasets
- Provide interesting baselines and insights

---

> ### Author Rebuttal · Authors · 2024-08-17
>
> We appreciate the reviewer’s suggestions for improving our manuscript, particularly regarding the discussion of limitations in our conclusion. We provide additional discussions with the following two aspects:
>
> - **RNA Tasks with Other Inputs**: To ensure fairness in comparison, we standardize input formats by using RNA sequences across all tasks and all models. This approach was necessary to maintain consistency in evaluations. However, it may not fully capture the potential of models that are designed to leverage richer input data, such as the scRNA-seq data for cell annotation and gene regulation network which uses genes with expression level [r1] and inverse RNA folding using structure as input [r2]. Future work could explore alternative evaluation settings that allow for the use of more complex inputs, thereby providing a more nuanced assessment of model performance.
>
> - **Other important RNA sequence tasks**: While our current benchmark effectively evaluates RNA language models across a spectrum of tasks, we acknowledge the out-of-selection of certain complex tasks such as 3D RNA structure prediction. This is primarily due to several challenges: (1) The majority of open-source RNA language models have not been developed to perform 3D structure prediction tasks. (2) The acquisition of high-quality structural data, often requiring Multiple Sequence Alignment (MSA) data, is prohibitively expensive. This, coupled with the limited availability of such data, poses a significant barrier [r3]. (3) The computational pipeline for predicting 3D structures (e.g., structural modules) is exceedingly complex and resource-intensive, often necessitating days of computation on high-performance GPUs [r4].
> In future extensions of our benchmark, we aim to incorporate more diverse tasks, including those requiring complex data inputs and significant computational resources, to better encompass the breadth of challenges in RNA research.
>
>  [r1] Theodoris, C. V., Xiao, L., Chopra, A., Chaffin, M. D., Al Sayed, Z. R., Hill, M. C., ... & Ellinor, P. T. (2023). Transfer learning enables predictions in network biology. Nature, 618(7965), 616-624.
>
>  [r2] Hofacker, I. L., Fontana, W., Stadler, P. F., Bonhoeffer, L. S., Tacker, M., & Schuster, P. (1994). Fast folding and comparison of RNA secondary structures. Monatshefte fur chemie, 125, 167-167.
>
>  [r3] Wang, W., Feng, C., Han, R., Wang, Z., Ye, L., Du, Z., ... & Yang, J. (2023). trRosettaRNA: automated prediction of RNA 3D structure with transformer network. Nature Communications, 14(1), 7266.
>
>  [r4] Li, Y., Zhang, C., Feng, C., Pearce, R., Lydia Freddolino, P., & Zhang, Y. (2023). Integrating end-to-end learning with deep geometrical potentials for ab initio RNA structure prediction. Nature Communications, 14(1), 5745.

---

### Author Response · Authors · 2024-08-25
**Thank all the reviewers for your time and efforts! Looking forward to the feedback.**

Dear Reviewers,

Thank you all for your time and efforts in reviewing our paper. We sincerely appreciate your constructive comments and thorough evaluation, which have been invaluable in helping us further enhance the quality of our work.

In response to your initial feedback, we have carefully prepared and submitted detailed responses to each of you. We are eager to hear your thoughts on whether our revisions have effectively addressed your concerns. If you have any questions, feel free to discuss them.

Once again, thank you for your valuable suggestions and the time you have dedicated to this process. We look forward to receiving your feedback.

---

### Decision · Program_Chairs · 2024-09-26

**Decision:**

Accept (Poster)

**Comment:**

In this work, the authors aggregate 13 distinct RNA-relevant tasks, and introduce the first RNA-focused benchmark effort. This is an area of growing scientific importance, and crucially, dataset standardization and harmonization remains a significant challenge in biology. All reviewers agreed the paper should be accepted and 3/4 placed it in the top 50% of all submissions. In particular, this work will facilitate the development of genomics-focused models, which often are hampered by an overreliance of techniques from the NLP community, where nature of the data is only superficially similar to genomics. I was particularly impressed with the thoroughness of the responses to authors and the incorporation of this information into the submission. While one author lamented the absence of an inverse RNA task, I agree with the authors that it is beyond the scope of this work (i.e., the work as submitted is already substantial enough) and could be added at a future date as a valuable extension of this benchmark. I also agree with the criticism that baselines are incomplete, but this is overcome by the utility of the benchmark itself, which I believe could be easily extended by the community after having reviewed the github repo. I would ask the authors to include all model details in the paper, this serves as a better single source of truth, rather than relying on github, both for discoverability and also in the case that the code is updated (your github repo has no official paper release, so could easily fall out of sync with the published paper). I trust you will prepare these changes for the camera ready version.